# Specific post-translational modifications of soluble tau protein distinguishes Alzheimer's disease and primary tauopathies

Nathalie Kyalu Ngoie Zola ®[1,2,9], Clémence Balty ®[2,9], Sébastien Pyr dit Ruys ®[3], Axelle A. T. Vanparys ®[1], Nicolas D. G. Huyghe ®[4], Gaëtan Herinckx ®[5], Manuel Johanns ®[2], Emilien Boyer[1,6], Pascal Kienlen-Campard ®[1], Mark H. Rider[2], Didier Vertommen ®[5] & Bernard J. Hanseeuw ®[1,6,7,8,10] ✉

Tau protein aggregates in several neurodegenerative disorders, referred to as tauopathies. The tau isoforms observed in *post mortem* human brain aggregates is used to classify tauopathies. However, distinguishing tauopathies *ante mortem* remains challenging, potentially due to differences between insoluble tau in aggregates and soluble tau in body fluids. Here, we demonstrated that tau isoforms differ between tauopathies in insoluble aggregates, but not in soluble brain extracts. We therefore characterized post-translational modifications of both the aggregated and the soluble tau protein obtained from *post mortem* human brain tissue of patients with Alzheimer's disease, cortico-basal degeneration, Pick's disease, and frontotemporal lobe degeneration. We found specific soluble signatures for each tauopathy and its specific aggregated tau isoforms: including ubiquitination on Lysine 369 for cortico-basal degeneration and acetylation on Lysine 311 for Pick's disease. These findings provide potential targets for future development of fluid-based biomarker assays able to distinguish tauopathies in vivo.

Tauopathies are a group of neurodegenerative diseases characterized by the accumulation of pathologically misfolded tau protein. Albeit being classified as a secondary tauopathy due to the association of tau pathology with amyloidosis, Alzheimer's disease (AD) is the most prevalent tauopathy amongst at least twenty others[1,2]. It has been established that tau pathology correlates more strongly with AD cognitive impairment than amyloidosis[3].

The definite differential diagnosis of tauopathies is based on *post mortem* neuropathological examination of the brain[4]. Besides histological differences in the type of tau aggregates, biochemical studies demonstrated differences in tau isoforms found in these aggregates[5]. Tau indeed exists as six isoforms in the adult human brain, and the number of repeats of the microtubule-binding domain defines 4R- and 3R-tau isoforms[6]. Under physiological conditions, the abundance of

[1]Universite catholique de Louvain (UCLouvain) and Institute of Neuroscience (IONS), 1200 Brussels, Belgium. [2]Universite catholique de Louvain (UCLouvain) and de Duve Institute (DDUV), Protein Phosphorylation (PHOS), 1200 Brussels, Belgium. [3]Universite catholique de Louvain (UClouvain) and Louvain Drug Research Institute (LDRI), Integrated Pharmacometrics, Pharmacogenomics and Pharmacokinetics Group (PMGK), 1200 Brussels, Belgium. [4]Université catholique de Louvain (UCLouvain) and Institut de Recherche Expérimentale et Clinique (IREC), 1200 Brussels, Belgium. [5]Universite catholique de Louvain (UCLouvain), de Duve Institute (DDUV), and MASSPROT Platform, 1200 Brussels, Belgium. [6]Cliniques universitaires Saint-Luc, Neurology Department, 1200 Brussels, Belgium. [7]Universite catholique de Louvain (UCLouvain), WELBIO department, WEL Research Institute, avenue Pasteur, 6, 1300 Wavre, Belgium. [8]Harvard Medical School, Massachusetts General Hospital, Department of Radiology, Gordon Center for Medical Imaging, Boston, MA, USA. [9]These authors contributed equally: Nathalie Kyalu Ngoie Zola, Clémence Balty. [10]These authors jointly supervised this work: Didier Vertommen, Bernard J. Hanseeuw. ✉e-mail: bernard.hanseeuw@uclouvain.be

4R- and 3R-tau isoforms is equivalent[7]. In AD brain, a combination of 4R- and 3R-tau isoforms are observed in aggregates. In contrast, corticobasal degeneration (CBD) and progressive supranuclear palsy (PSP) are characterized by aggregates containing 4R-tau only while Pick's disease (PiD) is characterized by aggregates containing only 3R-tau. Frontotemporal lobe degeneration (FTLD) cases due to tau pathology are often either 4R- only or 3R-only tauopathies[2,7].

The clinical diagnosis of tauopathies is based on cognitive symptoms[8]; however, these symptoms do not depend on the pathological alterations in the tau protein, but on the anatomy of the tau deposits[9]. Because the relationships between the biochemical alterations of tau and the specific brain regions in which these alterations occur are weak[10], clinical diagnoses are not always confirmed after *post mortem* pathological examination. This mismatch between the clinic and pathology makes it difficult to test drugs targeting specific biochemical mechanisms in patients who have been diagnosed clinically. Therefore, developing in vivo biomarkers reflecting biochemical alterations of tau protein is a key challenge in the development of precision therapies for tauopathies.

AD can be confirmed using cerebrospinal fluid (CSF) measures of amyloid and total tau concentrations, as well as tau Threonine 181 phosphorylation (P-T181) which is typically above physiological thresholds in AD patients[11–13]. However, CSF tau 4R and 3R isoforms measures have not allowed distinguishing between patients with primary tauopathies[14–16]. According to previous works, tau protein post-translational modifications (PTMs) are molecular events that could distinguish between tauopathies[17,18]. A recent study using a sensitive, selective, and unbiased liquid-chromatography tandem mass spectrometry (LC-MS/MS) method was used to analyse AD and control brain samples and unravelled an association between aggregated tau PTMs and Braak stages, suggesting a potential impact of specific PTMs on AD progression[19]. However, this study did not include patients with non-AD tauopathies. Besides, most of these studies focused on brain aggregates, which are hardly observable in biological fluids that can be collected in vivo, such as CSF or plasma. However, biomarker development would benefit from mass spectrometry analyses conducted on brain material where the tau protein is more abundant than in body fluids.

In this work, we considered that body fluids measures would be better approximated by soluble rather than insoluble human brain material, as previously suggested[20]. We compared the abundance ratio between 4R and 3R tau isoforms both in aggregates and soluble extracts from *post mortem* human brain tissue of individuals with AD and non-AD tauopathies and characterized tau PTMs using an unbiased LC-MS/MS approach. Here, we demonstrate that the reported 4R/3R tau isoforms ratios can only distinguish between tauopathies in aggregates, not in the soluble brain fractions. Second, we show that some of the PTMs observed in the soluble fraction of the brain appear to be specific for certain tauopathies and correlate with their corresponding 4R/3R aggregated tau isoforms ratios. These PTMs constitute potential biomarkers to distinguish between tauopathies in biological fluids and could foster the development of therapeutic strategies.

## Results

### Subjects and brain section sample characteristics
Subjects and samples characteristics are provided in Table 1. The study used human *post mortem* cerebral sections from the Netherlands Brain Bank (NBB), sampled from individuals for whom definitive clinicopathological diagnoses were AD (n = 15), CBD (n = 5), PiD (n = 5), FTLD (n = 10) and non-demented control participants (CTL, n = 5). These samples were selected based on (1) a *post mortem* delay of maximum 8 h before freezing, (2) a *post mortem* neuropathological analysis showing at least a Braak neurofibrillary tangle stage[9] of V for AD patients and of maximum II for CTL. We selected these cases as

representative of "4R and 3R" (AD), "4R" (CBD), "3R" (PiD) and "4R or 3R" (FTLD) tau pathologies. FTLD cases all had diagnosed tauopathy (cases without tauopathy[21] were excluded from the analyses). We sampled the inferior frontal gyrus because this region is presumably affected by tau aggregates in patients with PiD and FTLD pathologies, as well as in patients with AD at Braak stage V–VI, but not in Braak stage 0–II CTL brains. We also sampled the precentral gyrus for CBD subjects as tau pathology affects this region at an early stage of this disease. No significant differences were found amongst the five groups of subjects in terms of sex, brain weight, and *post mortem* delay (Table 1, Kruskal–Wallis test and Fischer exact test, P > 0.10). AD patients had a higher frequency of the *APOE4* allele than CTL subjects, and CBD, PiD, and FTLD subjects were significantly younger than AD and CTL individuals. AD subjects had a higher mean amyloid CERAD score than non-AD subjects. Total tau was measured using the microtubule-binding region (MTBR) (i.e., the sum of 3R and 4R tau isoforms absolute quantities in insoluble and soluble fractions, see Supplementary Tables 1 and 2, respectively). In brain soluble tau, no significant differences were found in absolute total tau levels. In contrast, lower quantities of aggregated tau were observed in controls compared with patients having tauopathies.

### Comparison between the composition in tau isoforms in soluble and insoluble brain extracts
We first aimed to compare the ratio of 4R- and 3R-tau isoforms between tauopathies both in soluble and insoluble brain fractions (see Fig. 1 for tau protein isoforms sequence and domains). Neocortical sarkosyl-insoluble and sarkosyl-soluble tau proteins were isolated from human *post mortem* samples of AD, CBD, FTLD, PiD and CTL subjects and the fractionation was confirmed by western blotting (Fig. 2). A SureQuant[22] targeted LC-MS/MS assay, using 4R- or 3R-tau isoform-specific AQUA-grade isotopically labelled peptides[23] (see Fig. 1 and Supplementary Table 3 for AQUA-peptide details) as internal standards allowed absolute quantification of endogenous 4R- and 3R-tau isoforms. The insoluble tau 4R/3R ratio classification (Fig. 3a) separated subjects in three groups, namely 4R-exclusive (all five CBD and four FTLD cases, including two with the P301L *MAPT* mutation (subjects 35 and 37)), 3R-exclusive (all five PiD and six FTLD cases) or both 4R- and 3R-tau isoform aggregates (AD and CTL cases), highlighting heterogeneity amongst the FTLD cases and justifying the subdivision into FTLD-4R and FTLD-3R groups across all subsequent analyses. By contrast, in soluble tau, 4R/3R ratio mapping (Fig. 3b) showed no differences in terms of 4R- and 3R-tau isoforms across subjects. Absolute levels of 4R- and 3R- tau isoforms were significantly different between tauopathies in tau aggregates (Fig. 3c, P = 0.0001), but not in soluble fraction (Fig. 3d, P > 0.15). Insoluble tau was not observed in CTL inferior frontal sections, confirming the Braak stage 0–II reported from the *post mortem* neuropathological classification (Table 1, Fig. 3c). Distinguishing tauopathies in soluble tau is important for developing fluid-based biomarkers, as soluble, more than insoluble proteins, can be detected in plasma or CSF. Because the well-known 4R/3R tau isoform imbalance in tauopathies was only observed in aggregates and not in soluble tau, we next investigated tau PTMs rather than protein isoforms abundance, aiming to observe soluble PTMs discriminating between primary tauopathies.

### Comparative study of soluble and insoluble tau PTMs in tauopathies
The most studied PTMs in tau pathology[24], namely serine (S) and threonine (T) phosphorylation (P-), lysine (K) ubiquitination (Ub-), acetylation (Ac-) and mono-methylation (Me-) modifications were identified and quantified following untargeted LC-MS-MS analyses of sarkosyl-insoluble and sarkosyl-soluble tau protein. In the entire cohort, we identified 20 PTMs in tau aggregates and 42 PTMs in soluble tau (Supplementary Table 4), including some pathology-related PTMs,

**Table 1 | Subjects and sample characteristics**

|  | CTL (n = 5) | AD (n = 15) | CBD (n = 5) | PiD (n = 5) | FTLD (n = 10) | P value |
|---|---|---|---|---|---|---|
| Age at death | 79 [72–87] | 70 [64–78] | 69 [58–73] | 62 [57–67] | 60 [54–66] | 0.01 |
| Sex | 60 | 53 | 60 | 0 | 70 | 0.15 |
| APOE genotype | 40–0–60 | 27–60–13 | 60–0–40 | 40–0–60 | 20–40–40 | 0.05 |
| Brain weight | 1157 [1010–1385] | 1034 [942–1120] | 1044 [950–1293] | 973 [872–1445] | 938 [833–1055] | 0.14 |
| *Post mortem* delay | 6 [4–8] | 5 [4–6] | 7 [5–50] | 6 [5–8] | 6 [5–7] | 0.15 |
| Braak stage | 0–II | V–VI |  |  |  | NA |
| Amyloid CERAD | 80–20–0–0–0 | 0–0–0–100–0 | 0–40–0–0–60 | 20–0–0–0–80 | 20–0–0–0–80 | <0.001 |
| Total soluble tau | 553 [28–1077] | 678 [528–802] | 683 [494–1000] | 570 [433–866] | 647 [448–873] | 0.94 |
| Total insoluble tau | 9 [5–13] | 236 [147–482] | 193 [117–333] | 44 [19–79] | 64 [28–117] | 0.0 |

Values are shown as percentage or median with confidence intervals 95%. Age (years), brain weight (g), *post mortem* delay (hours between death and autopsy) and total tau values were analysed using two-tailed Kruskal–Wallis test (P < 0.05). Sex (% female), APOE genotype (% ε3–% ε4–% unknown), and amyloid CERAD scores (% 0 - % A - % B - % C - % unknow) were analysed using two-tailed Fischer exact test (P < 0.05). CBD, PiD, and FTLD were younger than AD and CTL subjects. AD cases were more frequently APOE ε4 carriers and had an amyloid CERAD score of "C" whereas none of the non-AD cases had such score. Braak stages (0–VI) between AD and CTL subjects were not compared as the cases were selected on that base. The absolute quantities of total tau (fmol) measured using the tau MTBR (3R + 4R isoforms, quantified by the Termo Fisher Scientific SureQUANT method) in soluble brain extracts did not differ between conditions, but was significantly elevated in disease groups compared to controls in insoluble tau (limit of quantification: 50 attomoles). *AD* Alzheimer's disease, *CBD* corticobasal degeneration, *PiD* Pick's disease, *FTLD* frontotemporal lobe degeneration, *CTL* control individuals, *CERAD* Consortium to Establish a Registry for Alzheimer's Disease, *APOE* apolipoprotein E, *NA* not applicable. Source data are provided as a Source Data file (see sheet 4 for sample complete information).

i.e., the ones that were significantly elevated in at least one tauopathy compared to CTL subjects (13 in tau aggregates and 19 in soluble tau, see Table 2). The frequencies and abundances of these PTMs are illustrated in Fig. 4. The sequence coverage of the full-length tau protein (2N4R isoform) by the identified peptides was 72% for tau aggregates (non-covered sequence portions: residue positions [25–87], [127–163], [195–209], [291–294], and [439–441]) and 91% for soluble tau (non-covered sequence portions: residue positions [1–5], [127–143], [164–170], [241–242], and [291–298]). Most of the tau pathology-related PTMs in aggregates (Fig. 4a) were restricted to the second proline-rich region, the first, third, and fourth MTBR, and the carboxy-terminal domain. In contrast, soluble PTMs emerged earlier in the sequence of the tau protein, from the first proline-rich region (Fig. 4b). In both aggregated and soluble tau, no pathology-related PTMs were observed in the splicing-sensitive MTBR2. The abundances of pathology-related PTMs observed in aggregated and soluble tau were then compared between tauopathies (Table 2, Wilcoxon comparison). We classified tau pathology-related PTMs in three categories namely (1) 4R-specific (PTMs mostly observed in tauopathies with 4R-tau aggregates; Table 2), (2) 3R-specific (PTMs mostly observed in tauopathies with 3R-tau aggregates; Table 2) and (3) AD-specific (PTMs exclusively observed in AD; Table 2).

In tau aggregates (Table 2, upper panel), Ub-K369 was exclusively observed in participants with CBD. Ac-K311 was classified as 3R-specific since only observed in AD (86% of the cases, see Fig. 4 for the PTMs frequency), FTLD-3R (67%), and a minority (40%) of PiD cases, as well as Me-K331, which was observed in FTLD-3R (67%) and the exact same PiD subjects (40%). Of note, Me-K331 was not observed in AD. Finally, Ub-K311 and P-S262 were classified as AD-specific because observed in all AD cases (100%), and in none of the other tauopathies.

In soluble tau (Table 2, bottom panel), Ub-K369 ranked among 4R-specific PTMs since observed in all CBD cases (100%), in 50% of the FTLD-4R subjects, and in a minority (20%) of AD cases while completely absent in individuals with PiD or FTLD-3R. Ub-K343 was observed in all CBD, FTLD-4R, and PiD cases (100%), in most AD subjects (67%), and in CTL (40%), but in none of the FTLD-3R subjects. The abundance of these 4R-specific PTMs was greater in CBD than in any other group (Fig. 5a–left). Ac-K311 appeared as 3R-specific in soluble tau, as it was observed in AD (60%), FTLD-3R (83%), and PiD (80%) subjects, but not in any case with CBD or FTLD-4R (Fig. 5a–right). Lastly, Ub-K311 (100%), Ub-K317 (87%), and Ub-K267 when coupled with P-S262 (87%) were exclusively observed in AD soluble tau.

## Correlation between soluble tau PTMs and the 4R/3R ratio in tau aggregates

We next correlated the abundance of all 42 PTMs observed in soluble tau with the insoluble 4R/3R tau isoform ratio to determine whether some soluble PTMs could indicate the isoforms present in tau aggregates (Fig. 5b). Without correction for multiple comparisons, 13 PTMs significantly correlated with the 4R/3R ratio, five 4R- and eight 3R-associated (Spearman r > 0.31, P < 0.05). After Bonferroni-correction, two PTMs, Ub-K369 and Ub-K343, strongly correlated with 4R-tau aggregates and two other PTMs, Ac-K311, and P-S184 + P-S185, strongly correlated with 3R-tau aggregates (Spearman r > 0.49, P < 0.001).

## Soluble Tau PTMs distinguishing Alzheimer's disease from primary tauopathies

As AD cases had both 4R- and 3R-isoforms in their aggregates (Fig. 3c), they also had 4R- and 3R-specific PTMs to a variable extent (Fig. 5a). However, Ub-K311, Ub-K317, and Ub-K267 + P-S262 were AD-specific (Fig. 6a). In contrast, phosphorylation sites that discriminated AD from CTL subjects (P-T212 + P-T217, P-T231 + P-S238 ± P-S237, P-S396 + P-S400 ± P-S404) were not AD-specific, as they were also observed in non-AD tauopathies (Fig. 6b). Of note, P-S262 alone (without Ub-K267) was elevated in CBD and FTLD; but in AD, P-S262 was systematically associated to Ub-K267 in soluble tau, making the combination of Ub-K267 and P-S262 a PTM event specific for AD (Fig. 6a and Fig. 4). These discrepancies between insoluble and soluble tau suggest a role of specific PTMs in the mechanisms of tau aggregation (see Fig. 7 for a PTMs comparison between insoluble and soluble tau), such that Ub-K267 might prevent aggregation when AD tau is phosphorylated on S262. P-T217 was also systematically associated with P-T212 in soluble tau; but P-T217 was observed alone, without P-T212, in tau aggregates, suggesting a protective effect of P-T212 against tau aggregation. Similarly, P-T231 was systematically associated with P-S237 (and sometimes P-S238) in AD soluble tau, but P-T231 was observed alone in tau aggregates; and P-S396 was systematically associated with P-S400 (and sometimes P-S404), but P-S396 was observed alone in tau aggregates, suggesting a protective effect of these additional sites of phosphorylation against tau aggregation.

Taken together, our results indicate that specific soluble tau PTMs are specifically associated with 3R-tau aggregates (Ac-K311, P-S184 + P-S185), 4R-tau aggregates (Ub-K343, Ub-K369), or AD-tau aggregates (Ub-K311, Ub-K317, Ub-K267 + P-S262), making

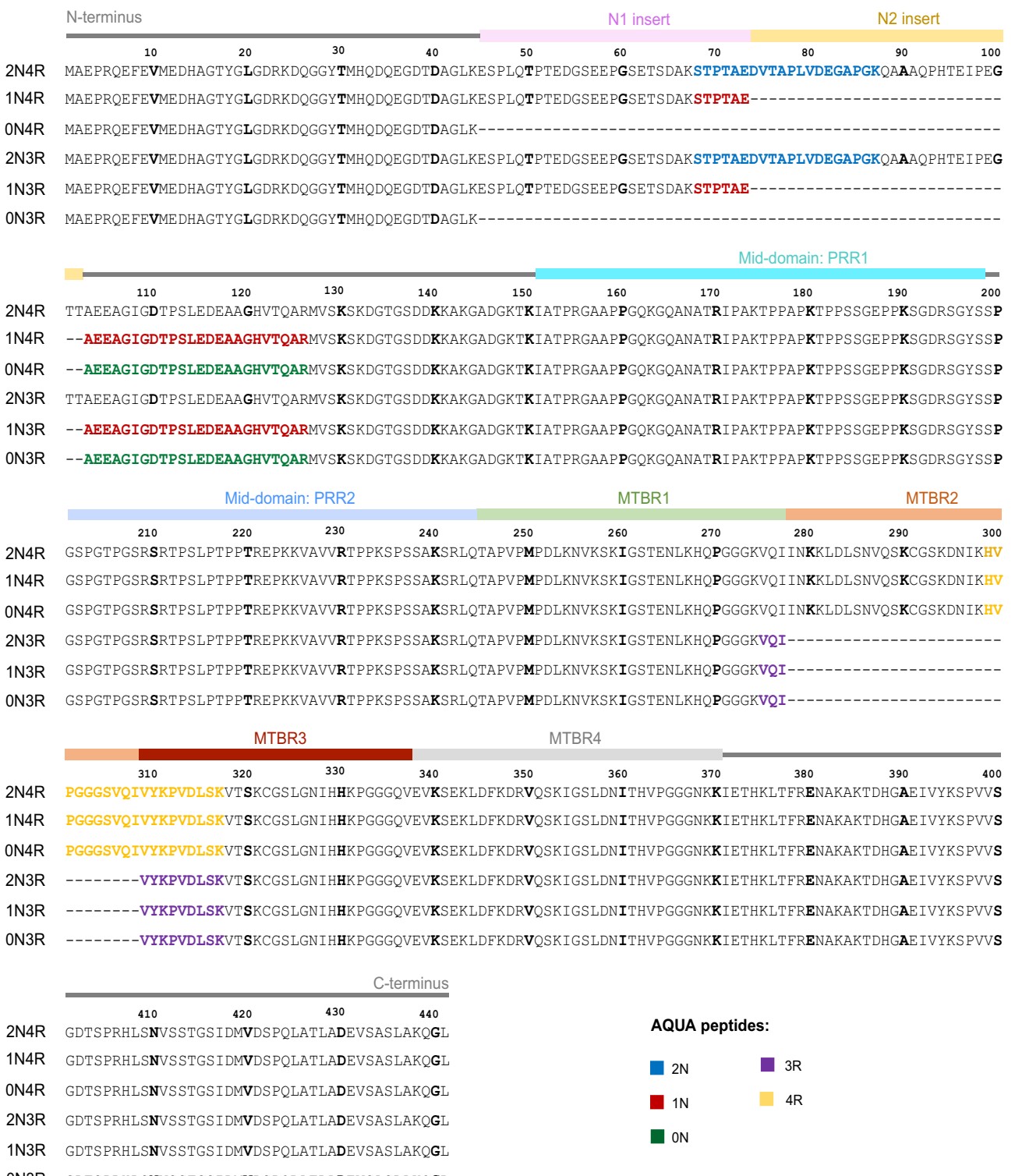

**Fig. 1 | Tau protein sequence and domains.** Amino-acid sequence of the six tau isoforms found in human central nervous system and generated by alternative splicing of *MAPT* gene exon 2 (encoding for N1 insert), exon 3 (encoding for N2 insert), and/or exon 10 (encoding for MTBR2). Sequences of AQUA-grade isotopically labelled peptides ("AQUA peptides") used for tau isoform absolute quantification are highlighted in colour. The AQUA peptides specific for the tau microtubule binding repeats were used for measuring the absolute quantities of total tau (sum of 4R and 3R tau isoforms) in the soluble and insoluble brain extracts. N1 and N2, tau protein projection domain N-terminal insert 1 and 2 respectively. PRR Proline-rich region, MTBR microtubule-binding repeat.

these soluble tau PTMs potential candidates for tauopathy distinction in biological fluids. The comparison between PTMs and the combination of PTMs into PTM events in insoluble and soluble tau provide indications on the role that PTMs play in the process of tau aggregation.

## Discussion

Tauopathies are classified *post mortem* by the type of isoforms of the tau protein aggregating in the human brain[7,17,25–27]. However, developing in vivo biomarkers allowing the distinction between tauopathies remains an unmet challenge[15]. To facilitate the translation of

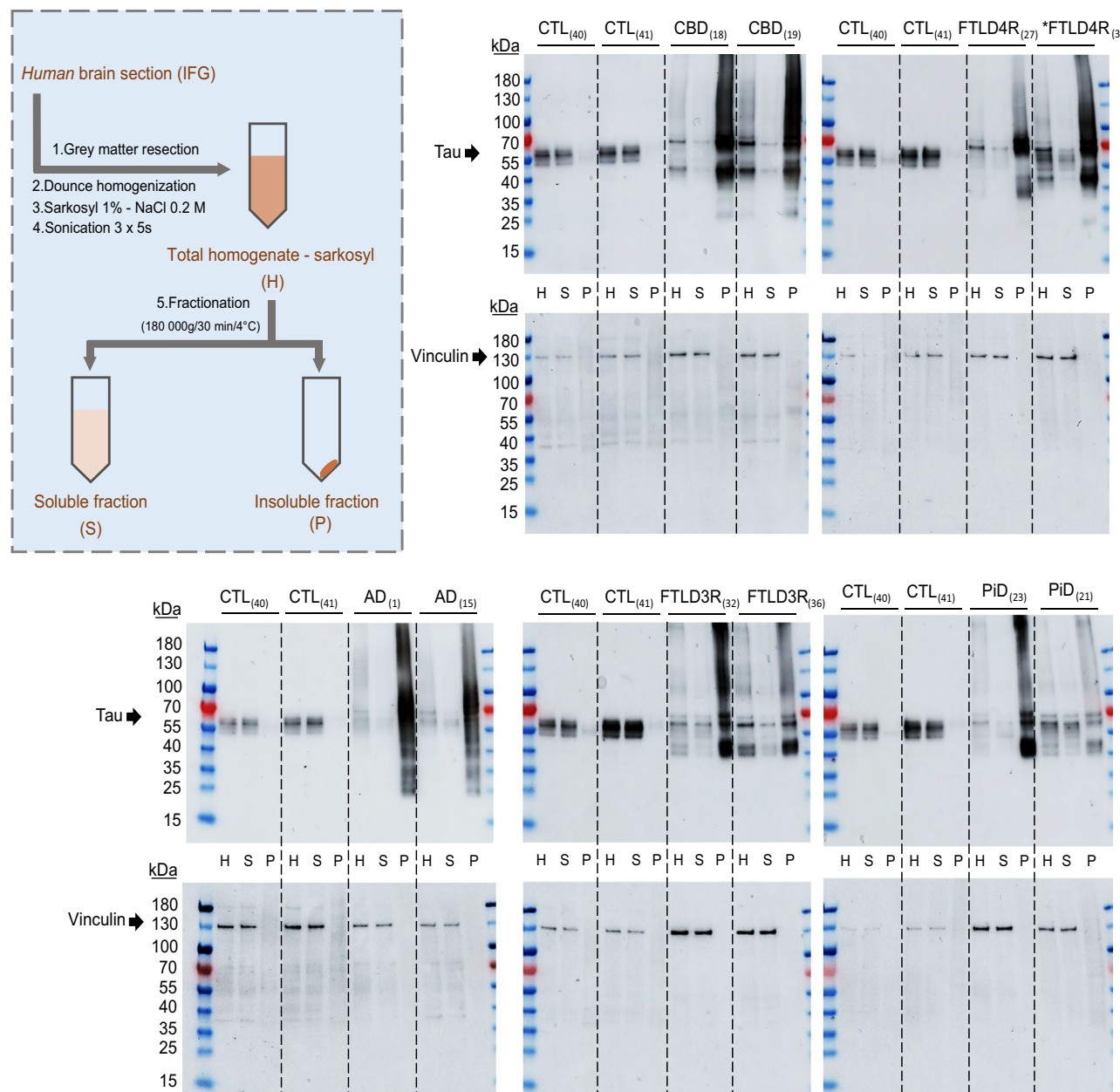

**Fig. 2 | Soluble and insoluble tau fractionation from total brain homogenate.** To validate ultracentrifugation-based tau fractionation from sarkosyl lysates, 5 µg of proteins from each fraction (H, total sarkosyl homogenate; S, soluble; P, insoluble pellet) from two subjects per tauopathy were analysed by western blotting using an antibody recognizing a stable epitope in Tau C-terminal domain. Vinculin was used as loading control for H and S fractions. AD Alzheimer's disease, CBD corticobasal dementia, FTLD frontotemporal dementia, PiD Pick's disease. Asterisks indicate a *MAPT* P301L mutation FTLD carriers. kDa kilodalton.

*post mortem* pathological observations into clinical biomarkers, we conducted a comparison of the human *post mortem* brain soluble and insoluble tau proteins, by performing targeted MS absolute quantification. We observed that the typical differences in isoform composition between tauopathies were exclusively found in tau aggregates, not in the tau protein extracted from the soluble fraction of the brain. As one may expect that CSF would mostly contain soluble rather than aggregated tau, this observation may explain the difficulty in distinguishing tauopathies using tau isoform analyses in CSF[15,16]. Besides, a recent MS study reported that the pattern of tau phosphorylation observed in CSF was more like the one observed in brain-soluble extracts than in aggregated tau[20]. We, therefore, used human *post mortem* brain samples and conducted a thorough characterization of the most studied tau PTMs, namely ubiquitination, acetylation, and mono-methylation on Lysine residues, and phosphorylation on Serine and Threonine residues,

using an untargeted MS approach to identify PTMs on soluble tau that would distinguish between tauopathies. We observed that specific brain soluble tau PTMs discriminated 4R- (Ub-K369, Ub-K343) and 3R- (Ac-K311 and P-S184 + P-S185) tauopathies. The level of these PTMs in AD were intermediate, as expected from the mixed 4R/3R tau isoforms composition in tau aggregates from AD brains (Fig. 5a). We also found AD-specific soluble PTMs (Ub-K311, Ub-K317, Ub-K267 + P-S262). Previous work using LC-MS/MS on *post mortem* brain tissue focused on comparing AD and controls[19]. In this recent study[19], a large number of pathology-related PTMs was observed in AD (including the three Ub sites mentioned above, together with 14 other ones), but it could not be concluded which PTMs were specific for AD, as no subjects with non-AD tauopathies were included.

Our results indicate that targeting these three AD-specific ubiquitination sites may be preferable for developing AD-specific assays in

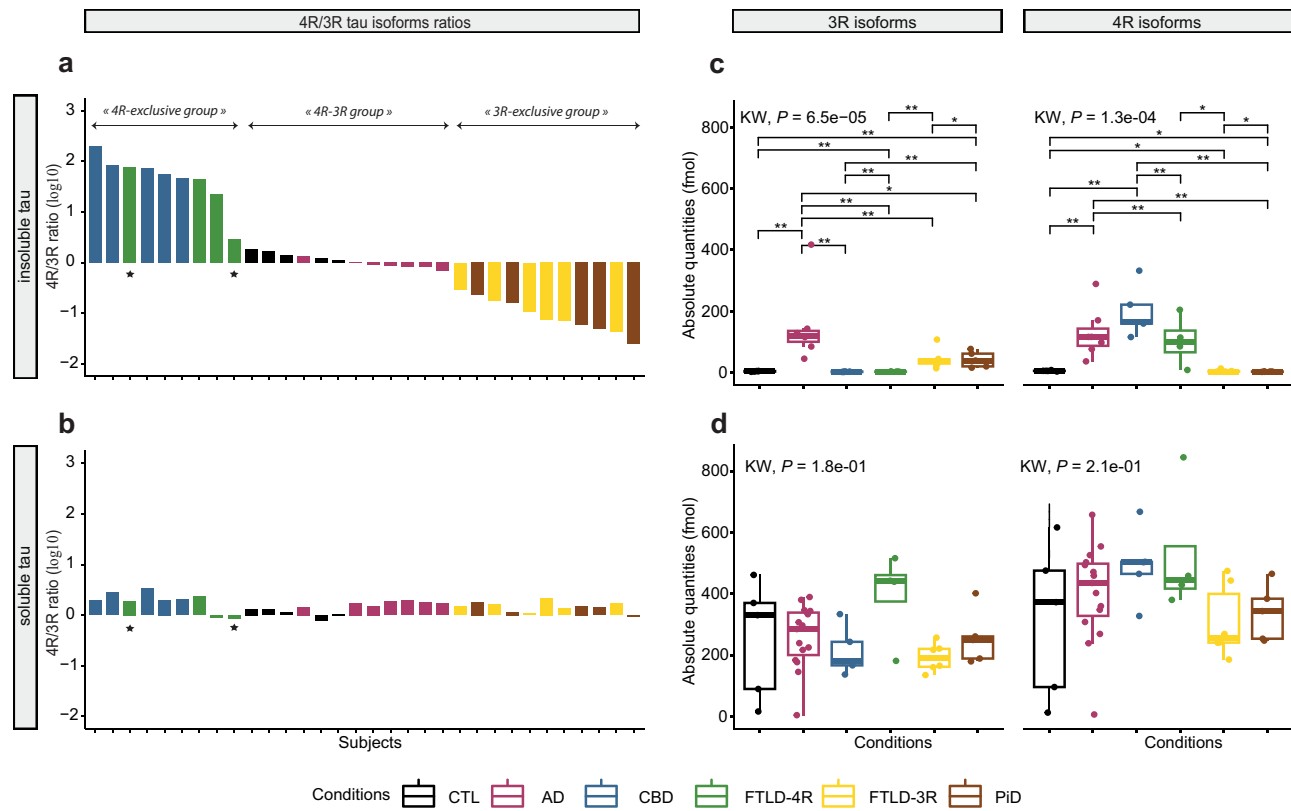

**Fig. 3 | Brain soluble and insoluble 4R and 3R tau isoforms ratios and distributions.** Left panels: plots of the log10 ratio between 4R and 3R tau isoforms in insoluble (**a**) and soluble (**b**) protein fractions from *post mortem* human brain with tauopathies. Each bar represents one subject from the highest (left) to the lowest (right) 4R/3R ratio observed in insoluble tau. A value of −1 means that this subject has ten times less 4R-tau isoforms than 3R-tau isoforms. **a** The 4R/3R ratio was greater in CBD and some FTLD cases (defined as FTLD-4R) than in AD and CTL cases, which was higher than in PiD and other FTLD cases (defined as FTLD-3R). Star symbols represent subjects with the *MAPT* P301L mutation. **b** In soluble tau, the 4R/3R ratio did neither differ between tauopathies, nor between *MAPT* P301L mutation carrier (star symbol) and non-carrier in the FTLD-4R group. Right panels: plots of the 3R and 4R tau isoforms absolute quantities in insoluble (**c**) and soluble (**d**) protein fractions. Each box represents a different condition. The line inside the box denotes the median value (50th percentile), while the box contains the 25th to 75th percentiles of dataset. The whiskers mark the 5th and 95th percentiles, and values beyond these upper and lower bounds are considered outliers. **c** Tau aggregates contained 3R-isoforms (left plot) exclusively in AD, FTLD-3R, and PiD and 4R-isoforms (right plot) only in AD, CBD, and FTLD-4R. **d** Soluble tau 3R and 4R protein isoforms did not differ between tauopathies. Data were analysed through multiple group comparisons using Kruskal–Wallis test (KW, *P* < 0.05), followed by a pairwise group comparison using an unpaired Wilcoxon test (**P* < 0.05, ***P* < 0.005, ****P* < 0.0005). All statistical tests were two-tailed. AD, Alzheimer's disease (*n* = 15 biologically independent samples); CBD corticobasal degeneration (*n* = 5 biologically independent samples); PiD, Pick's disease (*n* = 5 biologically independent samples); FTLD, frontotemporal lobe degeneration (*n* = 10 biologically independent samples, including FTLD-4R, *n* = 4 and FTLD-3R, *n* = 6); control individuals, CTL (*n* = 5 biologically independent samples). **a**, **b** The aggregates of eight AD individuals are not displayed for graphical reasons. Source data are provided as a Source Data file (see sheet 2 for the 4R/3R tau isoforms absolute quantities per subject).

CSF or plasma samples (Fig. 6a). Our study included brains from patients with AD and primary tauopathies. Four previous LC-MS-MS work also analysed non-AD brain tissue: one study only included PSP cases[28], another did not investigate PTMs[29], making a comparison with our study difficult. A third study showed 4R-specific changes only in CBD (but not PSP) and 3R/4R mixed tauopathies[30]. A fourth study observed Ub-K369 and Ub-K343 in aggregated tau from three CBD and three FTLD-4R cases with *MAPT* mutation, as well as Ac-K311 in PiD cases, which we all observed in our cohort, but this study did not analyse soluble tau extracts[18]. They did not observe differences between tauopathies in P-S185 (±P-S184), neither did we in the aggregates. There is thus consistent evidence that Ub-K369 and Ub-K343 are observed in 4R-tauopathies and Ac-K311 in 3R-tauopathies. We demonstrated these PTMs are not only detected in tau aggregates, but also in soluble tau extracts, making them suitable targets to develop fluid-based biomarker assays. We also showed that these soluble PTMs were associated with the ratio of tau isoforms in brain aggregates. Of note, no phosphorylation site was found specific for AD, although soluble tau P-T217, P-T231 and P-S396 distinguished AD from controls in our data, as reported previously in CSF and plasma[31–34], *post mortem* human brain tissue[35,36], and animal brain tissue[37]. We did not observe

pathological hyperphosphorylation of P-T181 in soluble tau, as this site was also phosphorylated in CTLs. The use of tau P-T181 in bio-fluid assays may therefore not be optimal compared to other AD-PTMs identified in this and other works[32] that should be investigated in vivo in future works. Because phosphorylation was also observed in non-AD tauopathies, assays targeting tau phospho-sites may not be optimal for AD differential diagnosis.

Although most previous research focused on tau phospho-sites as potential biomarkers or therapeutic targets[38–40], some recent works highlighted the importance of other types of PTMs in tauopathies[18,41,42], as ubiquitination or acetylation. A previous study identified the ubiquitin thioesterase Otub1 as a tau deubiquitinating enzyme, which activity enhanced the formation of tau aggregates in mice[43], suggesting that tau ubiquitination might be protective against aggregation, as our data suggested for Ub-K267. Another study showed that Ub-K311 and Ub-K317 deubiquitination increased tau acetylation and aggregation[44]. Comparing tau PTMs in the soluble and aggregated fractions of the brain allows identifying PTMs associated with tau aggregation or with the absence of aggregation (Fig. 7). Our findings are consistent with the view that P-T217, P-T231, P-S262, and P-S396 facilitate tau aggregation, as previously shown in the literature[36,45–47],

**Table 2 | Tau PTMs**

| | Kruskal–Wallis FDR adjusted P value | Wilcoxon group pairwise comparison | | | | | | | | | | | | | | |
|---|---|---|---|---|---|---|---|---|---|---|---|---|---|---|---|---|
| | | AD vs CTL | CBD vs CTL | FTLD 4R vs CTL | FTLD 3R vs CTL | PiD vs CTL | AD vs CBD | AD vs FTLD 4R | AD vs FTLD 3R | AD vs PiD | CBD vs FTLD 4R | CBD vs FTLD 3R | CBD vs PiD | FTLD3R vs FTLD4R | FTLD4R vs PiD | FTLD3R vs PiD |
| **Sarkosyl-insoluble tau (aggregates)** | | | | | | | | | | | | | | | | |
| Ub-K311 | 2E-04 | ++ | | | | | ++ | ++ | ++ | ++ | | | | | | |
| P-S262 | **1E-03** | ++ | | | | | ++ | + | ++ | ++ | | + | | | | |
| P-T217 | 4E-03 | ++ | ++ | | | | | + | ++ | ++ | | ++ | + | | | |
| P-S396 | 4E-03 | + | | | | | + | | + | + | | | | | | |
| P-S324 | 4E-03 | ++ | ++ | + | + | | | + | ++ | ++ | | ++ | + | | | |
| P-T231 + P-S238 | 8E-03 | ++ | ++ | + | | | | ++ | ++ | ++ | + | ++ | + | | | |
| Ub-K369 | **9E-03** | + | + | | | | - | | + | | | ++ | + | | | |
| Ac-K343 | 9E-03 | + | ++ | | | | | | + | | | ++ | + | | | |
| Ac-K353 | 2E-02 | + | | | | | | | | + | | | | | | |
| Ac-K311 | 2E-02 | + | | | + | | + | + | + | | | - | | | | |
| Ac-K369 | 4E-02 | + | | | | | | | + | + | | | | | | |
| P-T231 | 4E-02 | ++ | ++ | + | + | + | | + | + | + | | | + | | | |
| Me-K331 | **4E-02** | + | | | + | | | | | | | - | | | | |
| **Sarkosyl-soluble tau** | | | | | | | | | | | | | | | | |
| Ub-K254 | 4E-04 | ++ | ++ | + | ++ | + | | ++ | ++ | ++ | + | ++ | ++ | | | |
| Ub-K311 | **4E-04** | ++ | ++ | + | ++ | + | ++ | ++ | ++ | ++ | ++ | ++ | | | | |
| Ub-K259 | 4E-04 | ++ | + | + | ++ | - | | + | ++ | ++ | + | ++ | ++ | | + | |
| P-S262 | 4E-04 | ++ | ++ | + | ++ | | --- | - | --- | | | | | | | |
| P-S262 + Ub-K267 | **4E-04** | ++ | ++ | + | ++ | + | ++ | + | ++ | ++ | | | | | | |
| Ub-K317 | **4E-04** | ++ | ++ | + | ++ | | + | ++ | ++ | ++ | | | | | | |
| Ub-K267 | 4E-04 | ++ | + | + | ++ | + | + | | ++ | ++ | ++ | ++ | ++ | | | |
| Ub-K369 | **1E-03** | ++ | ++ | + | ++ | | --- | | | | + | ++ | ++ | | | |
| Ub-K343 | **1E-02** | + | + | + | ++ | | -- | | + | | + | ++ | ++ | - | | - |
| P-T231 + P-S237 + P-S238 | 2E-02 | ++ | ++ | + | ++ | + | | + | + | | + | | | | | |
| P-S396 + P-S400 + P-S404 | 2E-02 | ++ | ++ | + | ++ | | | | | | + | | + | | | |
| Ac-K311 | **3E-02** | + | + | + | + | + | + | | | | | - | - | + | - | |
| P-S184 + P-S185 | 3E-02 | ++ | + | + | + | + | | | | | | | | | - | |
| P-S113 | 4E-02 | + | + | + | ++ | | | | | | | | | + | | |
| P-T212 + P-T217 | 4E-02 | ++ | ++ | | ++ | | | | - | | | | | | | |
| Me-K180 | 4E-02 | ++ | + | + | + | + | | | | | | | | | | |
| P-S396 + P-S400 | 4E-02 | + | + | + | ++ | + | | | | | | | + | | | |
| P-S185 + P-S191 | 5E-02 | + | ++ | + | ++ | + | | | | | | | | | | |
| P-T231 + P-S238 | 5E-02 | ++ | + | + | | | | | | | | | | | | |

Only pathology-related PTMs are displayed, i.e. the ones whose abundances (AUC) were significantly higher in a disease group than in CTL (FDR adjusted Kruskal–Wallis's P < 0.05). The table shows the result of the post-hoc Wilcoxon's pairwise group comparisons on PTMs (+/- P < 0.05; ++/-- P < 0.005; +++/--- P < 0.0005). All statistical tests were two-tailed. Source data are provided as a Source Data file (see sheet 5 for the exact P values). Plus sign and Minus sign represent either a significant increase or significant decrease, respectively, in the condition at the top position of the panel heading. PTMs with p-values in bold font are those specific for AD (insoluble tau Ub-K311, Ub-K317 and P-S262; soluble tau Ub-K311, Ub-K317 and P-S262+Ub-K267), for 4R-tauopathies (insoluble tau Ub-K369; soluble tau Ub-K343 and Ub-K369) and for 3R-tauopathies (insoluble tau Ac-K311 and Me-K331; soluble tau Ac-K311). PTMs are numbered according the full-length (2N4R) tau protein. AD Alzheimer's disease (n = 15), CBD corticobasal degeneration (n = 15), PiD Pick's disease (n = 5), FTLD frontotemporal lobe degeneration (n = 10, including FTLD-4R, n = 4 and FTLD-3R, n = 6); control subjects, CTL (n = 5).

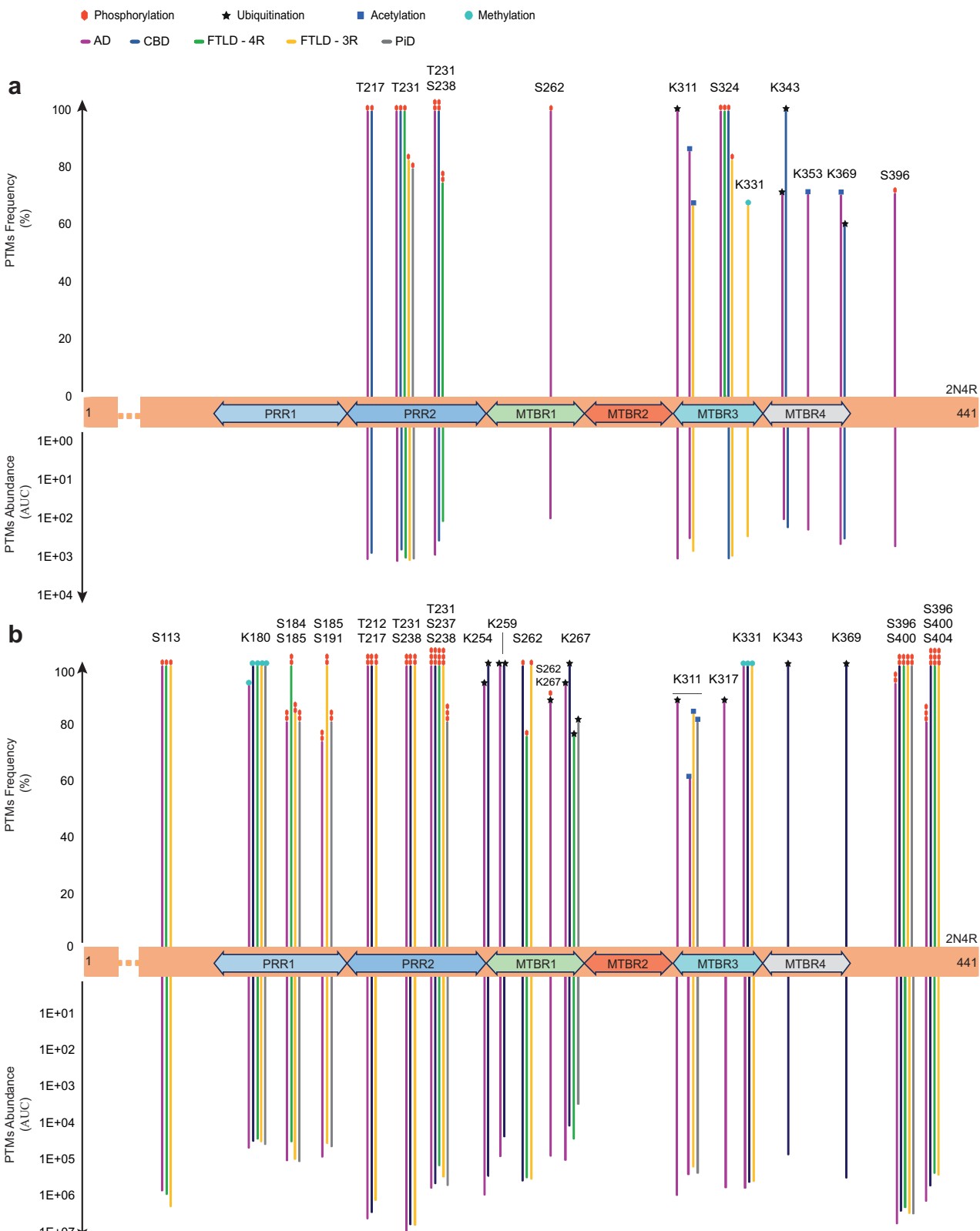

**Fig. 4 | Frequency and abundance of tau PTMs discriminating tauopathies from CTLs.** The occurrences of PTMs with significantly different abundances between any tauopathy and CTL are represented on the full-length tau protein sequence (2N4R, 441 amino acids) for the aggregated (a) and soluble (b) brain fractions. The dotted lines represent sequences where no pathology-related PTMs were found and omitted here for graphical reasons. The PTMs frequency is the proportion of subjects with a given tauopathy carrying the indicated modification; PTM abundances are areas under the curve (AUC) of corresponding mass spectrometry extracted ion chromatograms (XIC). Bar colours indicate the type of tauopathy, while symbol colours represent the type of PTMs (red polygon for phosphorylation, black star for ubiquitination, blue square for acetylation, and turquoise circle for mono-methylation). S Serine, K Lysine, T Threonine, PRR1 first proline-rich region (mid-domain), PRR2 second proline-rich region (mid-domain), MTBR microtubule-binding repeat, AD Alzheimer's disease, CBD corticobasal dementia, FTLD fronto-temporal dementia, PiD Pick's disease. See Table 2 for statistics. Source data are provided as a Source Data file (see sheet 3 for the abundances values and sheet 5 for the Wilcoxon test's exact *p* values).

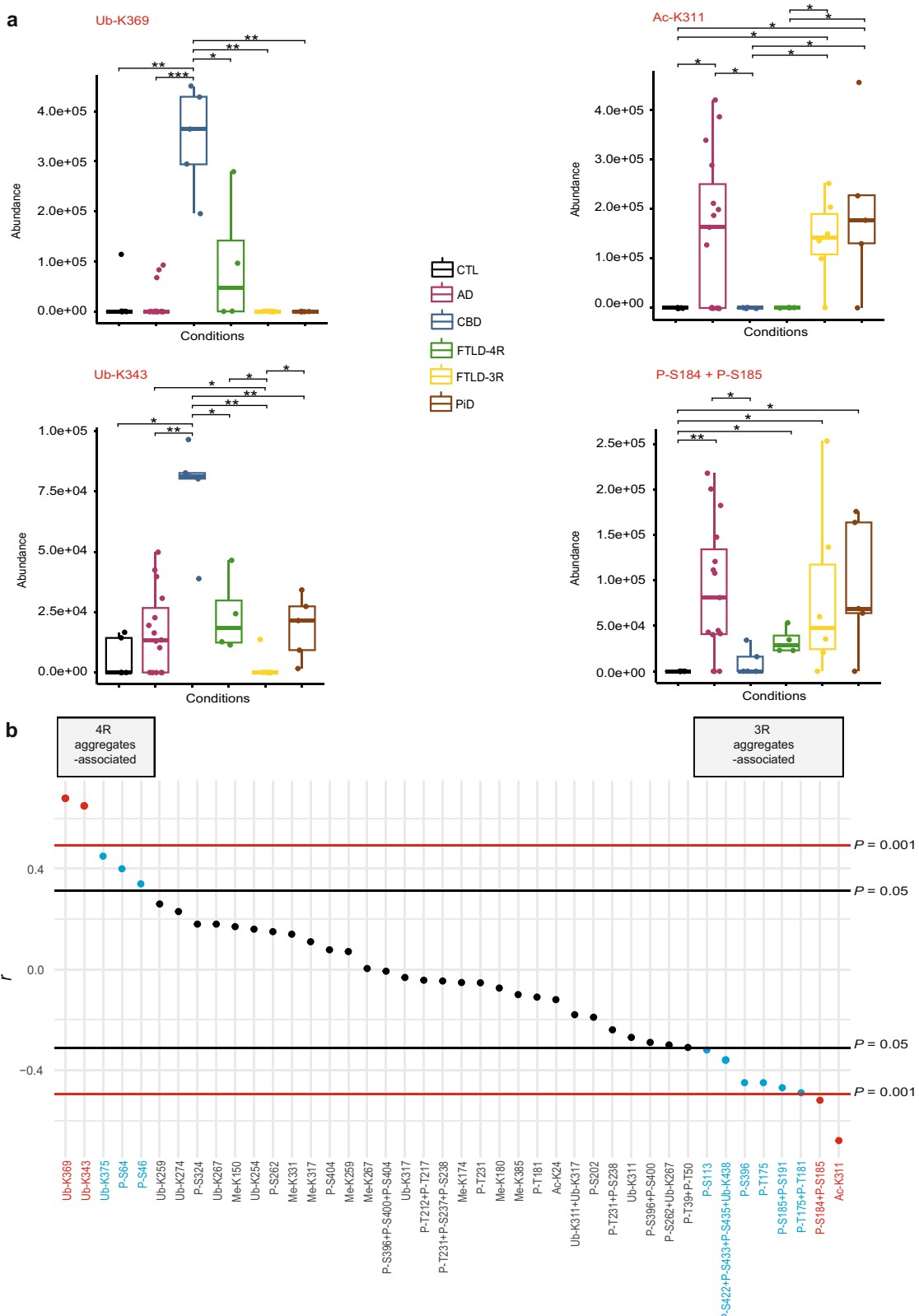

whereas P-T212, P-S237, Ub-K267, and P-S404 prevent tau aggregation[48], highlighting potential targets for the development of tauopathy treatments based on kinases, phosphatases, or ubiquitin signalling enzymes.

Our study does not come without limitations. We only sampled one brain region from participants with confirmed, and therefore advanced, pathology. Our data is therefore not informative for early

diagnosis of (preclinical) tauopathy. Future mass spectrometry work could investigate earlier cases and different brain regions from AD and non-AD tauopathies to evaluate the spatial and temporal progression of PTMs in the brain. To date, such regional studies have been carried out only by imaging of aggregated tau from AD brains with different Braak stages[49] or by antibody-based study of oligomeric and detergent-soluble tau from controls and AD brains at different Braak

**Fig. 5 | Soluble Tau PTMs associated with aggregated tau 4R/3R ratios.** PTMs are numbered according to full-length (2N4R) tau protein. Each box represents a different condition. The line inside the box denotes the median value (50th percentile), while the box contains the 25th to 75th percentiles of dataset. The whiskers mark the 5th and 95th percentiles, and values beyond these upper and lower bounds are considered outliers. **a** Left: soluble tau Ub-K369 and Ub-K343 were more abundant in 4R-tauopathies (CBD, FTLD-4R, and a few AD cases) than in 3R-tauopathies (PiD and FTLD-3R). Right: Ac-K311 and P-S184, when combined to P-S185, were more abundant in 3R-tauopathies (and AD to a variable extent) than in 4R-tauopathies. Data were analysed through multiple group comparisons using Kruskal–Wallis test ($P < 0.05$), followed by a pairwise group comparison using an unpaired Wilcoxon test (*$P < 0.05$, **$P < 0.005$, ***$P < 0.0005$). All statistical tests were two-tailed. **b** Spearman correlation between the ratio of insoluble 4R/3R tau isoforms and soluble tau PTMs showed that soluble tau Ub-K369 and Ub-K343 where strongly

correlated with 4R-tau isoform aggregates and that Ac-K311 and P-S184 + P-S185 were strongly correlated with 3R-tau isoforms aggregates after correction for multiple comparisons. Red lines are thresholds corrected for multiple comparisons (Spearman $r > 0.49$, uncorrected $P < 0.001$, Bonferroni-corrected $P < 0.05$). Black lines represent an uncorrected threshold (Spearman $r > 0.31$, uncorrected $P < 0.05$). All $p$ values are two-tailed. S serine, T threonine, K lysine, Ub- ubiquitination, P- phosphorylation, Ac- acetylation, Me- mono-methylation. AD Alzheimer's disease ($n = 15$ biologically independent samples), CBD corticobasal degeneration ($n = 5$ biologically independent samples), PiD Pick's disease ($n = 5$ biologically independent samples), FTLD frontotemporal lobe degeneration ($n = 10$ biologically independent samples, including FTLD-4R, $n = 4$ and FTLD-3R, $n = 6$); CTL, control ($n = 5$ biologically independent samples). Source data are provided as a Source Data file (see sheet 2 for the 4R/3R tau isoforms absolute quantities per subjects, sheet 3 for the abundances values and sheet 5 for the Wilcoxon test's exact $p$ values).

stages[50]. In addition, we limited our study to one exemplative pathology of "4R and 3R" (AD), "4R" (CBD), "3R" (PiD), and "4R or 3R" (FTLD). Tauopathies include more than twenty different diseases, and structural differences can be found in aggregates including the same type of tau isoform (e.g., in CBD versus PSP)[27]. Therefore, it would be worth characterizing tau PTMs in additional tauopathies to confirm whether the observed PTMs are specific for the type of isoforms found in aggregates or whether those PTMs are rather specific for the diseases that we have selected. Specifically, future work should determine whether Ub-K369 and Ub-K343 are specific for CBD or also observed in other 4R tauopathies such as PSP. Finally, the present study used soluble brain extracts, hypothesizing closer association with CSF data than when analysing aggregated tau. This hypothesis remains to be demonstrated. Indeed, contrary to brain soluble tau, CSF tau is heavily fragmented according to a previous study thoroughly profiling CSF tau fragments[51]. In particular, CSF tau appears to be lacking a large C-terminal portion (residues 275 to 441, in full-length tau), which would hinder the use of specific PTMs located in this region as biomarkers. However, there is still debate on the exact tau C-terminal portion missing in the CSF[30,52]. Future studies should thus use sensitive targeted methods to test the PTMs identified in this work (e.g. Ub-K369) in CSF and/or plasma samples and evaluate whether these PTMs can effectively distinguish between primary tauopathies in vivo.

## Methods

### Ethics
All Material and Data collected by the Netherlands Brain Bank (NBB) are obtained on the basis of written informed consent and comply with the local ethical and legislative requirements (see "Ethical and legal declaration of the Netherlands Brain Bank" provided in Supplementary information).

### Brain tissue sample processing and protein extract preparation
Insoluble and soluble Tau proteins were extracted from human *post mortem* brain tissue samples and fractionated. Therefore and following a well-established protocol[53], 0.2 g of cortical brain sample was collected by dissection while still frozen, for each subject. The dissected tissue was diced into 2 × 2 mm pieces while thawing and homogenized in 1 ml of low salt buffer (LS buffer: 50 mM HEPES, pH 7.0, containing 250 mM sucrose, 1 mM ethylene diamine tetra acetic acid (EDTA) and protease inhibitor cocktail (Roche)) by -15 strokes in a 2 ml Dounce homogenizer. The total homogenate volume was adjusted at 2 ml and mixed with 1% (w/v) Sarkosyl and 200 mM NaCl (final concentrations). The Sarkosyl-homogenates (H) were sonicated 3 × 5 s pulses at 30% amplitude on ice with a microtip probe and transferred to a 4 mL ultracentrifuge tubes for centrifugation in a pre-cooled Beckman 50Ti fixed-angle rotor at 180,000 × $g$ for 30 min at 4 °C. The supernatants, which corresponded to Sarkosyl-soluble protein fractions (S) were transferred to clean tubes and the pellet (Sarkosyl-insoluble proteins fraction, P) was dislodged/washed twice in low salt buffer, gently dried

and solubilized by incubation in urea buffer (50 mM Tris-HCl pH 8.5, 8 M urea) for 30 min at 20 °C followed by brief (1 s) sonication at 20% amplitude. For all samples, protein concentration was measured using a bicinchoninic acid assay kit (Thermo Scientific™, #23225). "S" fractions typically contained about 10 μg/μl of protein while "P" fractions contained about 4 μg/μl of protein.

### Validation of soluble and insoluble tau fractionation from total homogenate-sarkosyl
Soluble and insoluble tau fractionation were validated by Western Blotting using total sarkosyl homogenate, soluble and insoluble fractions from two subjects of each studied group (Fig. 2). 5 μg of total proteins per sample were loaded on 4–15% polyacrylamide gradient gels (Bio-Rad # 4561086). Membranes were incubated overnight at 4 °C with *rabbit* monoclonal antibody D1M9X recognizing an epitope surrounding the aspartate 430 of tau protein C-terminal domain (D430, numbering according to the full-length tau, 2N4R) (Cell Signaling Technologies # 46687) at a dilution of 1:1000. Vinculin was used as loading control (Cytoskeleton Inc. # AVN01 (clone VIN-11-5), dilution 1:1000). Signals were obtained by ECL and detected with an Amersham ImageQuant 800 CCD camera. More details about the antibodies used are available in Supplementary Table 5.

### Soluble tau enrichment
For each sample, soluble tau proteins were enriched from the "S" fractions by immunoprecipitation (IP). Total Sarkosyl-soluble proteins (2×250 μg) were incubated on a rotating wheel overnight at 4 °C with 20 μl protein G sepharose beads (Sigma-Aldrich, # GE17-0618-01) previously coupled and DMP-crosslinked (Sigma-Aldrich # D8388) with 2 × 2.5 μg/20 μl beads of *mouse* monoclonal HJ8.7 anti-tau antibody (D. Holtzman at Washington University School of Medicine; Missouri; USA Cat# HJ8.7, RRID:AB_2721234 [http://antibodyregistry.org/AB_2721234]), see Supplementary Table 5 for details). The beads were then washed four times with phosphate-buffered saline (PBS) to eliminate unbound protein and minimize salt for subsequent MS analyses. The washing buffer was removed without drying the bead, and immunoprecipitated proteins (coming from 2 × 250 μg of total proteins) were eluted with 200 μl 0.1% (v/v) formic Acid (FA) pH 2.7. The eluate was divided into two equal parts in clean tubes. An aliquot of 100 μl was taken for absolute quantification by targeted MS of soluble tau protein isoforms (SureQuant). The second 100 μl aliquot was used for identification of tau PTMs by untargeted tandem mass spectrometry (MS/MS).

### Sample preparation for soluble tau Isoform absolute quantification (targeted MS)
In the absence of robust brain soluble tau phospho-site LC-MS/MS data, we considered that soluble tau isoforms could be phosphorylated at different sites, such that MS absolute quantification could be biased by an under-evaluation of phosphorylated peptides. Therefore, 100 μl of

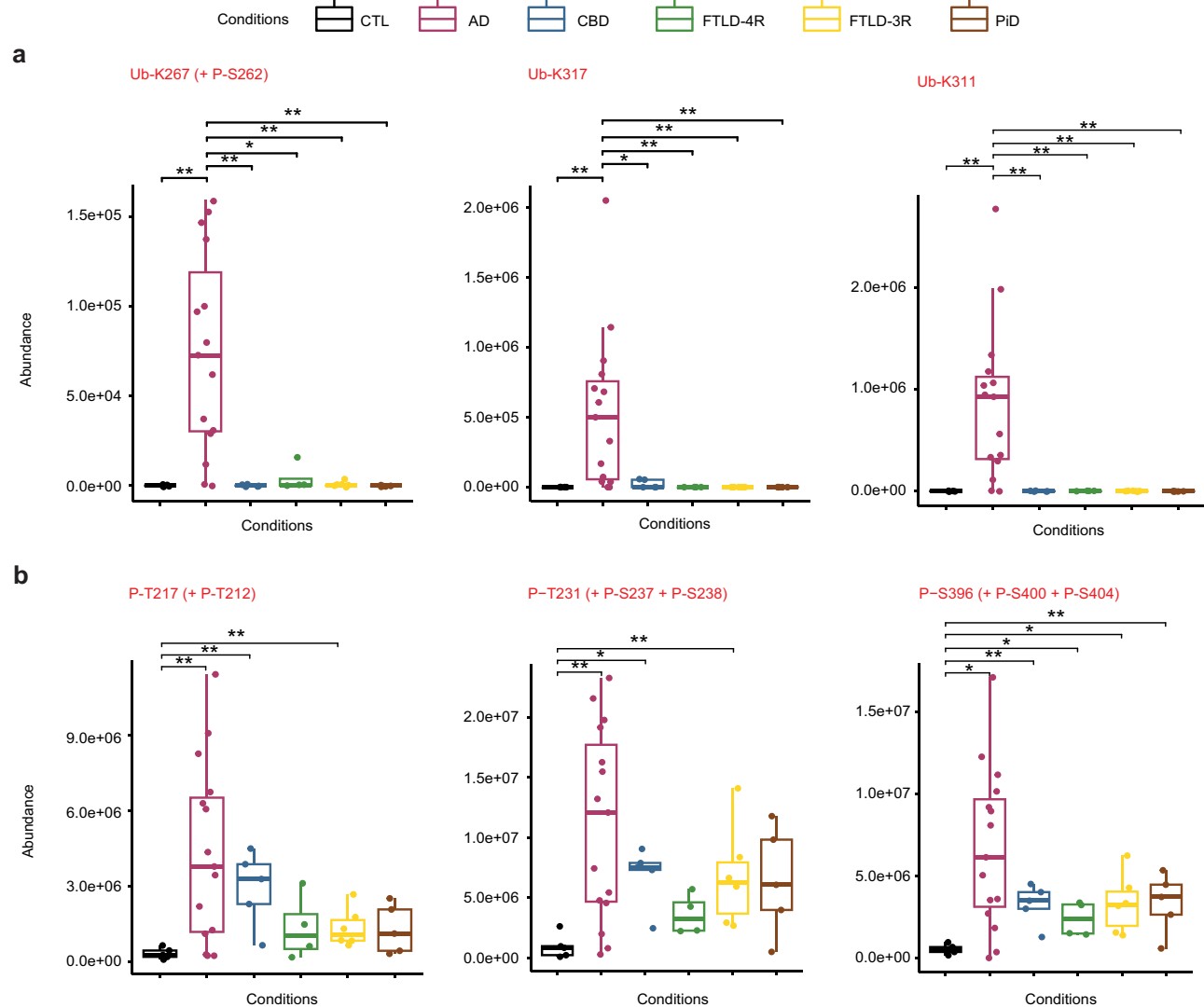

**Fig. 6 | Alzheimer's disease-specific and non-specific biomarkers in soluble tau.** Soluble tau PTMs specific to AD (a) and those not allowing to distinguish between AD and non-AD tauopathies (b) are numbered according to full-length (2N4R) tau protein. Each box represents a different condition. The line inside the box denotes the median value (50th percentile), while the box contains the 25th to 75th percentiles of PTM abundances (AUC of XIC). The whiskers mark the 5th and 95th percentiles, and values beyond these upper and lower bounds are considered outliers. **a** Soluble tau Ub-K267 (when associated with P-S262), Ub-K317, and Ub-K311 were exclusively identified in AD, not in CTL or other tauopathies. **b** Soluble tau P-T217 (when associated to P-T212), P-T231 (when associated to P-S238 with or without P-S237) and P-S396 (when associated to P-S400 with or without P-S404)

were significantly different from controls in all tauopathies but were not able to distinguish between AD and non-AD tauopathies. Data were analysed using an unpaired Wilcoxon test (*$P < 0.05$, **$P < 0.005$, ***$P < 0.0005$). All statistical tests were two-tailed. S serine, T threonine, K lysine, Ub- ubiquitination, P- phosphorylation. AD Alzheimer's disease ($n = 15$ biologically independent samples), CBD corticobasal degeneration ($n = 5$ biologically independent samples), PiD Pick's disease ($n = 5$ biologically independent samples); FTLD, frontotemporal lobe degeneration ($n = 10$ biologically independent samples, including FTLD-4R, $n = 4$ and FTLD-3R, $n = 6$); control individuals, CTL ($n = 5$ biologically independent samples). Source data are provided as a Source Data file (see sheet 3 for the abundances values and sheet 5 for the Wilcoxon test's exact $p$ values).

post-IP FA eluate was vacuum-dried and resuspended in volatile buffer (50 mM sodium carbonate-bicarbonate pH 9.9) for overnight agitation with 1 unit of agarose-bound alkaline phosphatase (Sigma-Aldrich, #P0762) at 37 °C. Following centrifugation ($400 \times g$ for 3 min at room temperature), the samples were vacuum-dried, resuspended in 40 µl buffer (100 mM triethylammonium bicarbonate, pH 8.5), and digested with 2 ng/µl of sequencing grade trypsin (Promega, #V511A) at 37 °C for 16 h in a thermomixer (Eppendorf) without previous alkylation step. The samples were vacuum-dried and resuspended in 10 µl MS injection buffer (3.5% (v/v) acetonitrile (ACN), 0.1% (v/v) trifluoroacetic acid (TFA)) containing 1 ng/µl of AQUA-grade isotopically labelled peptide specific for each tau isoform, which were obtained from Synpeptide Co Ltd (See Supplementary Tables 3 and 6 for AQUA peptides details). Sample (1 µl) was injected for LC-MS/MS analyses.

### Sample preparation for soluble tau PTM identification (untargeted MS)
Post-IP FA eluate (100 µl) were vacuum-dried and resuspended in 40 µl buffer (100 mM triethylammonium bicarbonate, pH 8.5) for digestion with 2 ng/µl of sequencing grade trypsin (Promega, #V511A) at 37 °C for 16 h without previous alkylation. Digests were vacuum-dried and resuspended in 10 µl MS injection (3.5% (v/v) acetonitrile (ACN), 0.1% (v/v) trifluoroacetic acid (TFA)) and 1 µl of mixture was injected for LC-MS/MS analyses.

### Sample preparation for aggregated tau isoform absolute quantification and PTM identification
No enrichment step was performed on the insoluble protein fractions, since less than ten different proteins were shown to become insoluble

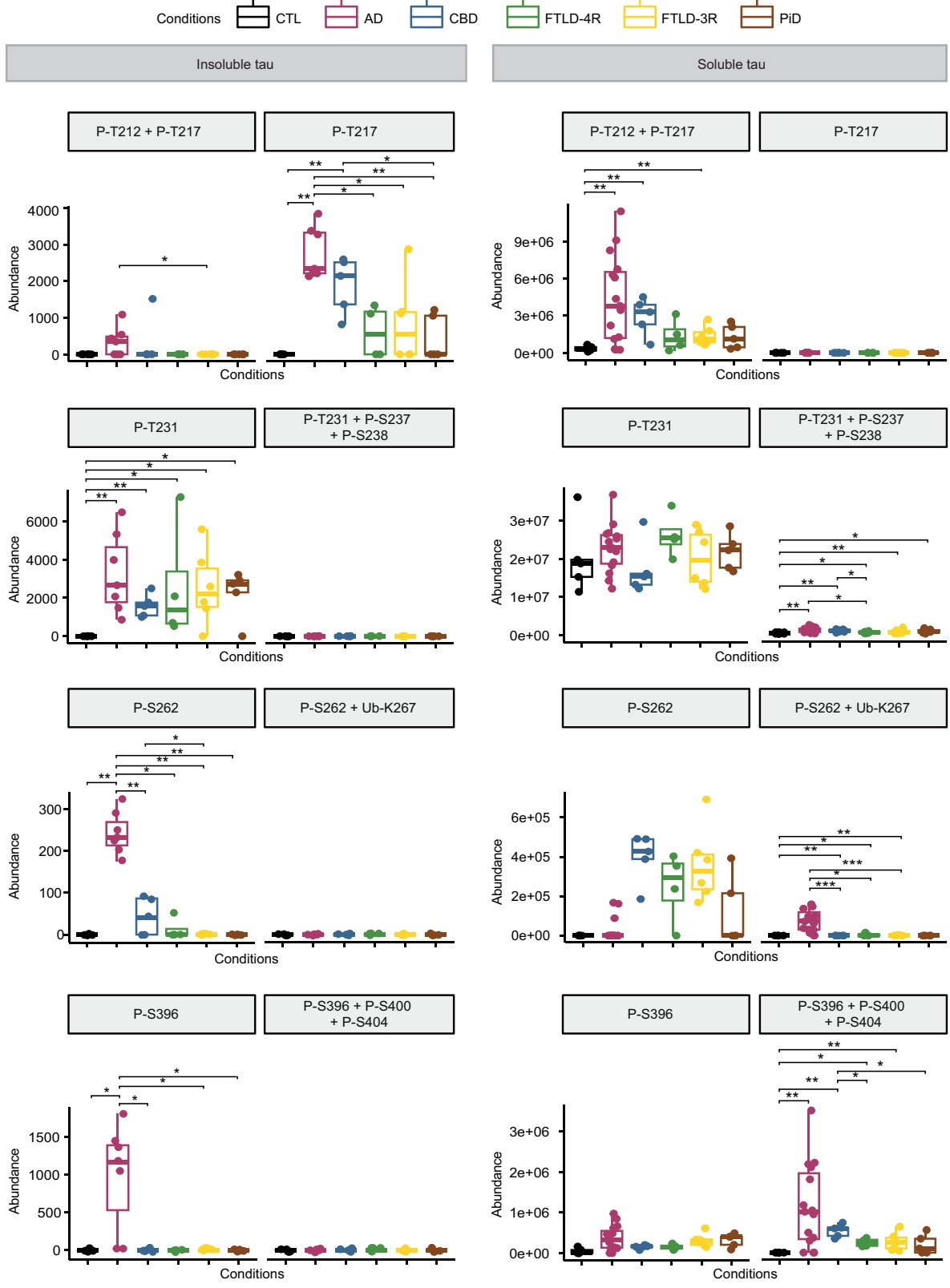

in brains with tauopathies[53]. Solubilized aggregates (P) were dilute 8-fold in 50 mM Tris-HCl pH 8.5 to decrease the urea concentration to 1 M final. Proteins (4 μg) were digested by incubation with sequencing grade trypsin (Promega, #V511A) at a trypsin:protein ratio of 1:50 at 37 °C for 16 h. The samples were vacuum-dried and resuspended in 10 μl MS injection buffer (see above) and the peptide concentration was measured using a quantitative colorimetric peptide assay kit (Thermo Scientific™, #23275). For aggregated tau isoform absolute

quantification, an aliquot of the peptide mixture (100 ng/μl) was transferred to a clean tube containing tau isoform-specific AQUA peptides (1 ng/μl final). No dephosphorylation step was done on tau aggregates since the endogenous sequences corresponding to 3R and 4R AQUA peptides (amino acids 306–317, according the numbering of 2N4R full-length tau) are virtually not phosphorylated in aggregates[54] excepted for P-S305, but this phosphorylation has been shown to inhibit tau aggregation[55].

**Fig. 7 | Tau post-translational modification (PTM) events.** PTMs are numbered according to full-length (2N4R) tau protein. Each box represents a different condition. The line inside the box denotes the median value (50th percentile), while the box contains the 25th to 75th percentiles of dataset. The whiskers mark the 5th and 95th percentiles, and values beyond these upper and lower bounds are considered outliers. PTMs are either observed alone or as PTM events, i.e., associated with other, nearby PTMs. The comparison between insoluble (left) and soluble (right) tau PTM events indicates that the combination of PTMs is more frequently observed in soluble than in insoluble tau, suggesting a protective effect of these additional PTMs against tau aggregation. Non-detected modified peptides were set arbitrarily to 0. The detection of the same peptide differently modified ensures that the absence of detection was not due to lack of ionisation or other methodological reasons. Data were analysed using an unpaired Wilcoxon test (*$P < 0.05$, **$P < 0.005$, ***$P < 0.0005$). All statistical tests were two-tailed. S serine, T threonine; K lysine; Ub- ubiquitination; P- phosphorylation. AD Alzheimer's disease ($n = 15$ biologically independent samples); CBD corticobasal degeneration ($n = 5$ biologically independent samples); PiD Pick's disease ($n = 5$ biologically independent samples); FTLD frontotemporal lobe degeneration ($n = 10$ biologically independent samples, including FTLD-4R, $n = 4$ and FTLD-3R, $n = 6$); control individuals, CTL ($n = 5$ biologically independent samples). Source data are provided as a Source Data file (see sheet 3 for the abundances values and sheet 5 for the Wilcoxon test's exact $p$ values).

## SureQuant quantification of recombinant tau

For LC-MS/MS data acquisition, the Thermo Scientific™ SureQuant method was used to set targeted peptide transition lists (Supplementary Table 6) for the absolute quantification of endogenous tau isoforms. For validation, this method was applied to known increasing amounts of tryptic peptides from the digestion of a recombinant unmodified full-length human Tau protein, kindly provided by Isabelle Landrieu (Institut Pasteur de Lille, France). Recombinant tau (1 μg) was subjected to methanol-chloroform precipitation and resuspended in buffer (100 mM triethylammonium bicarbonate, pH 8.5) for overnight digestion with 20 ng sequencing grade trypsin (Promega, #V511A). Vacuum-dried recombinant tau-peptides were reconstituted at 0.01 ng/μl, 0.1 ng/μl, 1 ng/μl, and 10 ng/μl in MS injection buffer (3.5% (v/v) acetonitrile (ACN), 0.1% (v/v) trifluoroacetic acid (TFA)) containing 1 ng/μl of each tau isoform-AQUA-peptide and 100 ng/μl of HeLa cell protein extract digest as matrix. A volume of 1 μl of each of the four mixtures was injected for LC-MS/MS analysis.

## LC-MS/MS absolute quantification and data analysis

Analysis was performed using the Orbitrap Fusion Lumos Tribrid Mass Spectrometer (Thermo Fisher Scientific) connected to a C18 peptide trap (300 μm × 5 mm) online with a PepMap C18 reversed-phase analytical column (75 μm x 250 mm) (Thermo Fisher Scientific). The sample was injected onto the peptide trap column at a flow rate of 20 μl/min in 3.5% (v/v) ACN/ 0.1% (v/v) TFA. After 3.5 min, the valve was switched and the peptides were eluted in backflush mode. Elution was performed in buffer B (80% (v/v) ACN / 0.1% (v/v) FA) using a multistep linear gradient of 4% to 22.5% B in 35 min, to 50% B in 20 min and finally to 100% B in 10 min. The flow rate (300 nl/min) was delivered by a RSLC nanopump Ultimate3000 (Thermo Fisher Scientific). The Orbitrap analyser was operated in PRM mode using a resolution of 120,000 for MS1 and 60,000 for targeted MS2 scans, an Automatic Gain Control (AGC) target of $10×10^5$ Ions and maximum injection times of 50 ms in MS1 and of $5 × 10^5$ and 116 ms in MS2. Peptides were selected for MS/MS fragmentation with HCD collision energy set at 32%, internal standard-triggered data acquisition using isolation windows of 1 Th and with minimal intensities pre-determined through the Thermo Fisher Scientific SureQuant method. This method enabled high-quality MS2 of target peptides ions due to a fragmentation trigger threshold set at 6 transition matches (PRM) between heavy AQUA experimental daughter-ions and theoretical counterparts (see Supplementary Table 6, for targeted mass trigger). All tau isoform-AQUA-PRM raw mass spectrometry files were analysed using Skyline software (see Supplementary Table 5 for details). Daughter ion intensities were extracted and light to heavy (L/H) ratios were define for each. Endogenous tau isoform quantities were defined as means of the ratios between daughter-ions. All extracted ion chromatograms were manually inspected. Pairwise group comparisons were performed for each tau isoform quantity via unpaired Wilcoxon's tests ($P < 0.05$) and multiple group comparisons were performed using the Kruskal–Wallis test ($P < 0.05$). All statistical tests were two-tailed.

## Untargeted LC-MS/MS identification and data analysis

Analysis was performed using the Orbitrap Fusion Lumos Tribrid Mass Spectrometer (Thermo Fisher Scientific) with the same chromatographic set up and MS configuration described above. Peptide ions were detected in the Orbitrap at a resolution of 120,000. Ions were selected for MS/MS using HCD setting at 30; ion fragments were detected in the Orbitrap at a resolution of 60,000. A data-dependent procedure that alternated between one MS scan followed by MS/MS scans was applied for 3 s for ions above a threshold ion count of 5.0E3 in the MS survey scan with 60.0 s dynamic exclusion. The electrospray voltage applied was 2.1 kV. MS1 spectra were obtained with an AGC target of 4E5 ions and a maximum injection time of 50 ms. MS2 spectra were acquired with an AGC target of 1E5 ions and a maximum injection time set to dynamic. For MS scans, the m/z scan range was 350 to 1800. The resulting MS/MS data were processed using the *Sequest HT* search engine within Proteome Discoverer (version 2.5, see Supplementary Table 5 for details) on a human protein database obtained from Uniprot (proteome ID: UP000005640). Trypsin was specified as cleavage enzyme allowing up to 2 missed cleavages, 4 modifications per peptide and up to 5 charges. Mass error was set to 10 ppm for precursor ions and fragment ions. Methionine oxidation (+15.995 Da), phosphorylation of serine, threonine, and tyrosine (+79.966 Da), ubiquitination of lysine (+114.042), acetylation and mono-methylation on lysine (+42.016 Da and +14.015 Da respectively) were considered as variable modifications. False discovery rate (FDR) was assessed using the program Percolator and thresholds for protein, peptide, and modification sites were specified at 1%. For abundance comparison, abundance ratios were calculated by Label Free Quantification (LFQ) of the precursor intensities within Proteome Discoverer 2.5 SP1. Modified (phosphorylation, ubiquitination, acetylation, and mono-methylation) peptide abundances corresponded to their respective extracted ion chromatogram. Pairwise group comparisons were performed for each PTM by unpaired Wilcoxon's tests ($P < 0.05$). For each PTM, multiple group comparisons were performed via Kruskal–Wallis test ($P < 0.05$) and a false discovery rate (FDR) with a threshold specified at 0.01. Where this latter test gave a significant difference among the compared group, a pairwise group comparisons were performed via unpaired Wilcoxon's tests ($P < 0.05$) All statistical tests were two-tailed. For the soluble fractions, as the abundances of total peptides and tau-peptides were constant across all samples (Fig. 8a), no normalization was applied. Despite constant amounts of total peptides across all insoluble fractions, amounts of tau-peptides were variable (Fig. 8b). Insoluble sample peptide abundance values were thus normalized to the tau R-domain (Fig. 8c).

## Statistics and reproducibility

No statistical method was used to predetermine sample size. For primary tauopathies, sample sizes were based on the availability of *post mortem* brain tissue ($n = 5$ PiD, $n = 5$ CBD, $n = 10$ FTLD-tau). We then selected 5 control cases and 15 AD cases to have a minimum of five individuals per group and provide sufficient variability in the ratio of 4R-/3R-tau isoforms. A minimal sample size of $n = 5$ per group was deemed sufficient as it allows observing significant differences

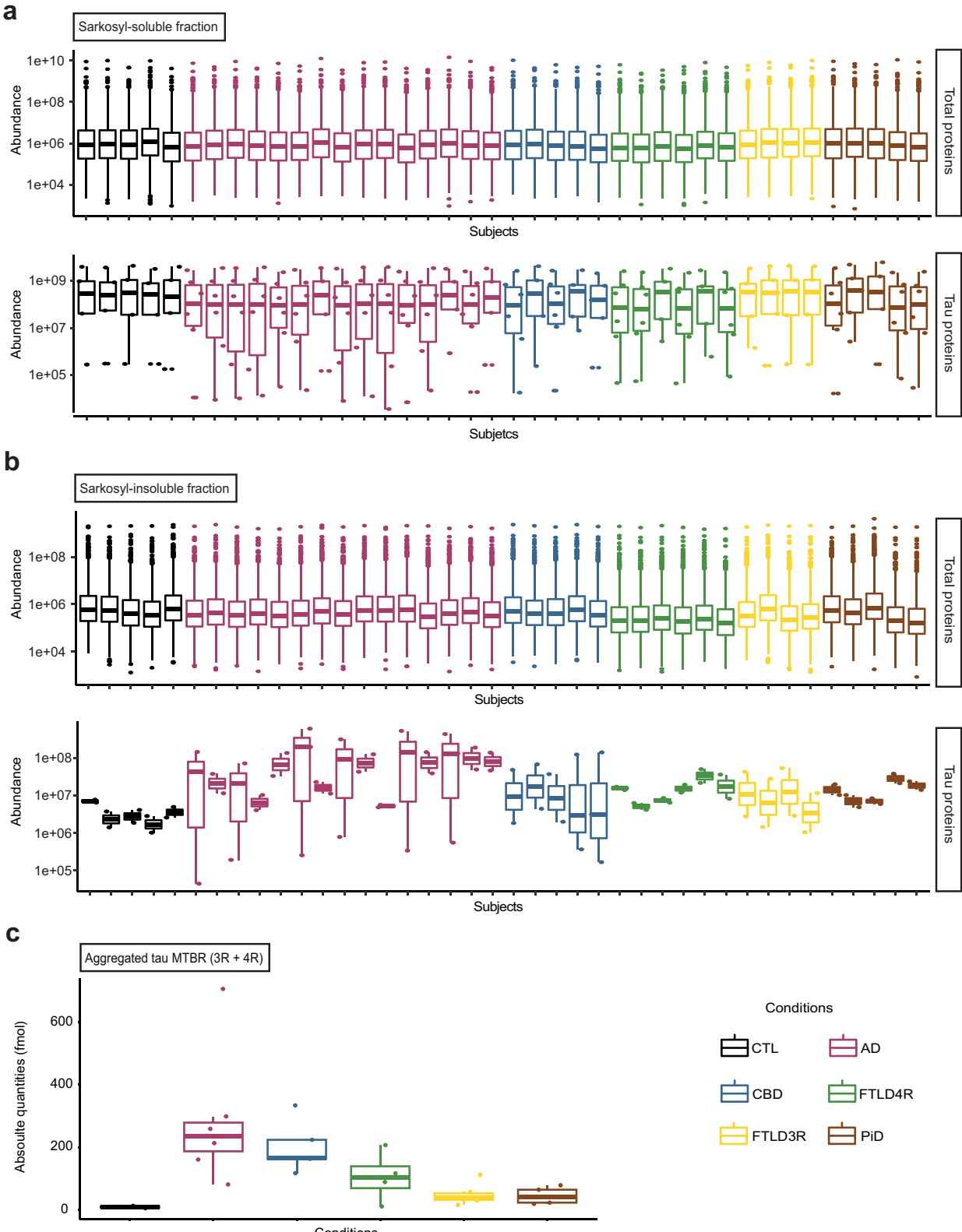

between groups for tau isoforms in the aggregates. A total sample of $n = 40$ was deemed sufficient as it allows disclosing correlations between PTMs and the ratio of 4R-/3R-tau isoforms when those variables share at least 10% of variance ($R^2 > 0.10$, $P < 0.05$). Data from sample 34 (FTLD original group) were excluded from all analyses for technical reasons (insufficient biological material). Data from samples 28, 33, and 38 (FTLD original group) were excluded from downstream analyses due to the impossibility of classification as either "4R-

exclusive" or "3R-exclusive" tauopathy based on their aggregates, which was required for this study. MS data were acquired once for each subject and each type of acquisition. Subjects were not allocated to randomized groups (this is not a clinical trial). Both in targeted (tau isoforms absolute quantities) and untargeted (tau PTMs) MS data, the studied groups were compared through a multiple group comparisons Kruskal–Wallis test ($P < 0.05$) and a false discovery rate (FDR) with a threshold specified at 0.01. For insoluble and soluble tau 4R-, 3R

**Fig. 8 | Sample total protein and tau protein mass spectrometry input. a** Soluble fraction proteins (top panel) and soluble tau (bottom panel) protein input were constant across all sample. **b** Aggregated proteins input was constant across all injected samples, but aggregated tau input varied by subject, justifying normalizing PTM abundances in the aggregates on the absolute quantity of aggregated tau, measured using the tau microtubule-binding repeat (MTBR, known to be the main part of tau sequence that aggregates). **c** Absolute levels of the tau protein Microtubule-binding region (MTBR) across all studied groups. In **a** and **b** each box represents a different subject ($n = 44$ biologically independent samples) and in

**c**, each box represents a different condition (AD Alzheimer's disease ($n = 7$ subjects), CBD corticobasal degeneration ($n = 5$ subjects), PiD Pick's disease ($n = 5$ subjects), FTLD frontotemporal lobe degeneration ($n = 10$ subjects, including FTLD-4R, $n = 4$ and FTLD-3R, $n = 6$), CTL control individuals ($n = 5$)). The line inside the box denotes the median value (50th percentile), while the box contains the 25th to 75th percentiles of dataset. The whiskers mark the 5th and 95th percentiles, and values beyond these upper and lower bounds are considered outliers. Data are available via ProteomeXchange with identifier PXD038901.

isoforms and PTMs, when the Kruskal–Wallis test gave a significant difference among the compared groups ($P < 0.05$), a pairwise group comparisons were performed via unpaired Wilcoxon's tests ($P < 0.05$). All statistical tests were two-tailed.

### Reporting summary
Further information on research design is available in the Nature Portfolio Reporting Summary linked to this article.

## Data availability
The untargeted mass spectrometry proteomics data have been deposited in the ProteomeXchange Consortium via the PRIDE[56] partner repository with the dataset identifier PXD038901 and 10.6019/PXD038901. The human proteome database used for untargeted mass spectrometry analyses is available on UniProt under the identifier UP000005640. The targeted MS proteomic (SureQuant absolute quantification) raw quantification data will be made accessible upon request due to incompatibilities with the template of the approved and recommended data repositories. The corresponding quantification results are available in Supplementary Tables 1 and 2. Source data are provided with this paper.

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

## Acknowledgements

The Belgian Fund for Scientific Research (FNRS) provided grants for B.J.H. (#CCL40010417), C.B. (#T.0208.20), and M.J. (#T.0008.15). This work was also supported by the Fonds de la Recherche Scientifique – FNRS (#OL J.0099.20), for the FRFS-WELBIO under Grant n° 40010035, the Queen Elizabeth Medical Foundation, the Belgian Alzheimer Research Foundation, and a Concerted Research Action (Brainbrush). The authors thank Isabelle Landrieu for providing us with recombinant tau protein, David Holtzman from Washington University for providing us with the HJ8.7 anti-tau antibody, and Lieven Desmet from UCLouvain for statistical advice. The authors also thank the Netherland Brain Bank for providing the brain samples that have been used in this study.

## Author contributions

N.K.N.Z.: visualization, conceptualization, methodology, data analyses, writing—original draft preparation, writing—review and editing. C.B.: visualization, conceptualization, methodology, post-acquisition proteomic data analyses, writing—review and editing. S.P.d.R.: visualization, methodology, writing—review and editing. A.A.T.V.: visualization, writing—review and editing. N.D.G.H.: data analyses, writing—review and editing. G. H.: methodology, writing—review and editing. M.J.: Methodology, writing—review and editing. E.B.: writing—review and editing. P.K.-C.: visualization, writing—review and editing. M.H.R.: resources, writing—review and editing. D.V.: visualization, Supervision, mass spectrometry proteomics analyses, writing—review and editing. B.J.H.: visualization, supervision, conceptualization, writing—original draft preparation, writing—review. and editing, project administration, funding acquisition. All authors have read and agreed to the published version of the manuscript.

## Competing interests

The authors declare no competing interests.
