## [Peer Review File · Nature Communications]

Specific post-translational modifications of the soluble tau protein distinguish between Alzheimer's disease and primary tauopathiesREVIEWER COMMENTS

Reviewer #1 (Remarks to the Author):

In the present manuscript, the authors analyze post-translational modification patterns on the Tau protein in different Tauopathies by quantitative mass spectrometry. They provide a solid overview of modification patterns in different diseases, grouped by 3R or 4R tauopathies, and further compare modification differences on soluble Tau species. Although mass spectrometry-based studies on Tau PTMs have been performed in the past, this study adds value, as it compares multiple tauopathies and examines modifications on both soluble and insoluble Tau. I therefore expect the data to be of high interest to the community.

Before the study can be considered for publication, I further recommend that the authors address the following points:

- Abstract: The study is put in the context of biomarker discovery, and its potential application to CSF biomarkers is highlighted. However, no CSF samples were analyzed in the present data set, and indeed the authors write in the discussion that translation of the findings to CSF needs to be demonstrated in future studies. I therefore recommend avoiding translatability statements in the abstract to avoid readers being misled.
- Translatability to CSF: Tau in the CSF is heavily fragmented, and a thorough profile of fragments that can be detected in the CSF has been published (Sato et al, Neuron, 2018). Since particularly the C-terminus (aa275-441) is missing in the CSF, PTMs in this region are expected to translate poorly to biomarkers. This point should be discussed.
- Abstract: The claim 'characterize all post-translational modifications' is an overstatement, since there are many PTMs that the authors did not investigate (arginine methylation, SUMOylation, lipid modifications, glycosylation...)
- Sequence coverage: The authors state that the sequence coverage for insoluble Tau was lower than for soluble Tau (72% vs 91%). Please include information in the manuscript about which regions of the protein were not covered, particularly for insoluble Tau, since by default PTMs in these regions could not have been detected.
- IP with N-terminal Tau antibody: This step was only performed for measuring soluble Tau, and thus introduces a bias towards non-truncated Tau species. Statements concluding that truncated species are only present in the aggregates (page 3) are therefore not warranted, since any species lacking an N-terminus would not have been detected in the soluble fractions.
- Fig.2: Here the authors show PTMs that were significantly different in disease samples compared to controls. However, information on all PTMs (also those that are not different) is not included in the manuscript, but would be important for people in the field and for comparison to published studies.
- Methods: based on the mass change given for the methylation search I assume only mono-methylation was used as a variable modification? Since di- and tri-methylation can also theoretically occur, I would suggest to correct the text to 'mono-methylation on lysine'.
- Methods: In the MS methods section, no alkylation step is mentioned. This common procedure to improve efficiency of the tryptic digest can also introduce false-positive signals mimicking ubiquitination if iodoacetamide is used (Nielsen et al, Nat. Meth. 2008). The authors should state whether no alkylation, iodoacetamide or another alkylating agent was used during sample preparation.
- Patient sample information: Please indicate, which cortical regions were used for patient samples. Did any patients have known Tau mutations?

- Table 1: The total Tau quantification data for insoluble Tau provided in this table do not seem to match the data provided in Extended data Fig. S3. Furthermore, the table legend states that total Tau was below the detection limit of quantification in the insoluble fraction, while the legend of Fig. S3 states that PTM abundances were normalized to absolute quantity of aggregated Tau.

- Extended Data Fig. S3c: Were any of the Tau levels measured here below the limit of quantification? If so, normalization to such values is inappropriate.

- Discussion: Tau PTMs in earlier disease stages, across brain regions and in soluble fractions have been assessed using antibody-based approaches (Ercan et al, Acta Neuropath Comm 2019). The statements in the bottom paragraph of page 8 may be modified by including this study.

- Will the MS data generated in this study be made available through a repository?

- Figure 1a+b: please label the y-axis in the left panels.

- Both MTBR and MTBD are used to describe the microtubule binding region. Please standardize.

Reviewer #2 (Remarks to the Author):

In the present manuscript, Zola, Balty et al use mass spectrometry on post-mortem tissue extracts to identify post-translational modifications to tau associated with tauopathies. This is an area of intense investigation in the field, but the present manuscript provides two areas of novelty: 1) highlighting differences between AD and non-AD tauopathies and 2) identifying specific PTMs in soluble and insoluble fractions. The authors show that 3R/4R levels are altered primarily in insoluble and not soluble fractions and identify some 4R-specific PTMs such as Ub-K369. The authors postulate that future work could attempt to identify these PTMs in biofluids such as plasma and CSF as potential disease biomarkers. Overall, the manuscript is interesting, novel and timely. Some points for revision:

- In the introduction, for clarity it is worth explaining that AD is a secondary tauopathy
- The most common 4R tauopathy is PSP, but this disease is not mentioned in the manuscript - this should be included in the introduction, and I also think the omission of PSP has an impact on data interpretation detailed below
- The statement "...while fronto-temporal lobe degeneration (FTLD) cases due to tau pathology are always either 4R-only or 3R-only tauopathies" is untrue, FTLD-tau can also have pathology comprised of both 3R and 4R isoforms (for example FTLD linked to MAPT mutations such as R406W have NFTs similar to in AD with both 3R and 4R isoforms)
- The authors use a group of FTLD-tau cases including at least some MAPT mutation carriers. I think it is important to detail these - and the specific mutations - in the table as the mutations will impact on tau isoform deposition. For example, if mutations impacting tau splicing were included these are highly likely to impact on tau in the soluble fraction too
- Western blots should be included to confirm that the soluble/insoluble fractionation preps have been successful - ideally for each disease group, as it is likely that different tau banding patterns will be observed (including truncated tau)- but at a minimum for control and AD to show successful fractionation
- The findings of 4R specific modifications are very interesting. The authors should include some PSP cases, as it is important to know whether these changes are common across 4R tauopathies or if they might be able to distinguish between FTLD/CBD/PSP

ANSWERS TO THE REVIEWERS' COMMENTS

We thank the reviewers for their constructive and helpful comments that we feel have significantly improved our manuscript. Please find below our answers to each of the points raised by the reviewers. Our responses are indicated in red below. Copy-pasted sections from the original manuscript are in italics with changes in red. We provide page numbers in bold.

To comply with Nature Communication formatting instructions, the Manuscript's sections were organized and completed accordingly. As 10 display items are authorized, some formerly called "Extended Data Fig. s" were converted to Manuscript figures and replaced in the main text. Of note, a new figure (Fig. 7) was added, illustrating the validation of soluble and insoluble proteins sarkosyl fractionation. Finally, Extended Table were converted to Supplementary Table, now located in Supplementary information section alongside newly added (upon Reviewers' request) Supplementary Table 3 (**insoluble** tau 4R/3R isoforms Targeted MS proteomic quantification results), Supplementary Table 4 (**soluble** tau 4R/3R isoforms Targeted MS proteomic quantification results) and Supplementary Table 5 (**all** identified Tau PTMs).

Extended Data Fig. s1 → Fig. 5

Extended Data Fig. s2 → Fig. 6

Extended Data Fig. s3 → Fig. 8

Extended Table s1 → Supplementary Table 1

Extended Table s2 → Supplementary Table 2

Extended Table s3 → Supplementary Table 6

Reviewer #1 (Remarks to the Author):

In the present manuscript, the authors analyze post-translational modification patterns on the Tau protein in different Tauopathies by quantitative mass spectrometry. They provide a solid overview of modification patterns in different diseases, grouped by 3R or 4R tauopathies, and further compare modification differences on soluble Tau species. Although mass spectrometry-based studies on Tau PTMs have been performed in the past, this study adds value, as it compares multiple tauopathies and examines modifications on both soluble and insoluble Tau. I therefore expect the data to be of high interest to the community.

Before the study can be considered for publication, I further recommend that the authors address the following points:

- Abstract: The study is put in the context of biomarker discovery, and its potential application to CSF biomarkers is highlighted. However, no CSF samples were analyzed in the present data set, and indeed the authors write in the discussion that translation of the findings to CSF needs to be demonstrated in future studies. I therefore recommend avoiding translatability statements in the abstract to avoid readers being misled.

Reply: We agree that our work did not include any CSF analyses that will be the focus of future work. We have therefore modified the abstract (**page 1**).

"We found specific soluble signatures predictive of each tauopathy and its peculiar type of aggregated tau isoforms. These findings provide potential targets for future development of ~~for developing~~ cerebrospinal fluid assays able to distinguish between tauopathies in vivo."

- Translatability to CSF: Tau in the CSF is heavily fragmented, and a thorough profile of fragments that can be detected in the CSF has been published (Sato et al, Neuron, 2018). Since particularly the C-terminus (aa275-441) is missing in the CSF, PTMs in this region are expected to translate poorly to biomarkers. This point should be discussed.

Reply: We have modified the discussion following the reviewer's suggestion. The text has been changed **on page 6** (Discussion):

“Finally, the present study used soluble brain extracts, hypothesizing closer association with CSF data than when analysing aggregated tau. This hypothesis, although reasonable, remains to be demonstrated. Indeed, contrary to brain soluble tau, CSF tau is heavily fragmented (Sato et al, Neuron, 2018). In particular, CSF tau appears to be lacking a large C-terminal portion (residues 275 to 441 in full-length tau), which would hinder the use of specific PTMs located in this region as biomarkers. However, there is still debate on the exact portion missing in CSF as C-terminal fragments (e.g. amino acids 275-290: Horie et al., Nature Medicine, 2022; amino acids 354-406: Horie et al., 2021) have been detected and quantified in CSF from AD and non-AD patients. Future studies should thus use highly sensitive targeted methods to test the PTMs identified in this work (e.g. Ub-K369) in CSF and/or plasma samples and evaluate whether these PTMs can effectively distinguish between tauopathies in vivo.”

- Abstract: The claim ‘characterize all post-translational modifications’ is an overstatement, since there are many PTMs that the authors did not investigate (arginine methylation, SUMOylation, lipid modifications, glycosylation...)

Reply: We agree and have modified the abstract accordingly (page 1):

“We therefore used untargeted mass spectrometry to characterize ~~all~~ post-translational modifications of both the aggregated and the soluble tau protein obtained from human brain tissue of patients with Alzheimer’s disease, cortico-basal degeneration, Pick’s disease, and fronto-temporal lobe degeneration.”

- Sequence coverage: The authors state that the sequence coverage for insoluble Tau was lower than for soluble Tau (72% vs 91%). Please include information in the manuscript about which regions of the protein were not covered, particularly for insoluble Tau, since by default PTMs in these regions could not have been detected.

Reply: Thank you. The text has been modified to include the information (page 3):

“The sequence coverage of the full-length tau protein (2N4R isoform) by the identified peptides was 72% for tau aggregates (non-covered sequence portions: residue positions [25-87], [127-163], [195-209], [291-294] and [439-441]) and 91% for soluble tau (non-covered sequence portions: residue positions [1-5], [127-143], [164-170], [241-242] and [291-298])”

- IP with N-terminal Tau antibody: This step was only performed for measuring soluble Tau, and thus introduces a bias towards non-truncated Tau species. Statements concluding that truncated species are only present in the aggregates (page 3) are therefore not warranted, since any species lacking an N-terminus would not have been detected in the soluble fractions.

Reply: We agree and thank the reviewer for pointing this out. The inaccurate statement has been removed (page 3).

- Fig.2: Here the authors show PTMs that were significantly different in disease samples compared to controls. However, information on all PTMs (also those that are not different) is not included in the manuscript, but would be important for people in the field and for comparison to published studies.

Reply: We agree and have now included the extra information. A full table with all PTMs identified in insoluble and soluble Tau has been added as Supplementary Table 5, in Supplementary information section.

- Methods: based on the mass change given for the methylation search I assume only mono-methylation was used as a variable modification? Since di- and tri-methylation can also theoretically occur, I would suggest to correct the text to ‘mono-methylation on lysine’.

Reply: Thank you. The term “methylation” has been replaced by “mono-methylation” throughout the manuscript.

- Methods: In the MS methods section, no alkylation step is mentioned. This common procedure to improve efficiency of the tryptic digest can also introduce false-positive signals mimicking ubiquitination if iodoacetamide is used (Nielsen et al, Nat. Meth. 2008). The authors should state whether no alkylation, iodoacetamide or another alkylating agent was used during sample preparation.

Reply: We thank the reviewer for this relevant comment. Indeed, to avoid potential interference with ubiquitination signatures, no alkylation step was applied before trypsin digestion during sample preparation. This is now clearly stated in the methods section of the manuscript (**page 7**).

- Patient sample information: Please indicate, which cortical regions were used for patient samples. Did any patients have known Tau mutations?

Reply: As stated in the “Subjects and sample characteristics” part of the Result section (**page 2**), the cortical regions used were the Inferior frontal gyrus (IFG) for CTL, AD, FTLD and PiD subjects, and the precentral gyrus (PCG) for CBD individuals.

The reviewer has raised an interesting point about tau mutations. We now mention in the Results section (**page 2**) that Subjects 35 and 37 from the “FTLD-4R” group (High 4R/3R tau isoform ratio) carried the MAPT P301L mutation:

“The insoluble tau 4R/3R ratio classification (Fig. 1a) separated subjects in three groups, namely 4R-exclusive (all five CBD and four FTLD cases, including two with the P301L MAPT mutation (subjects 35 and 37)), 3R-exclusive (all five PiD and six FTLD cases) or both 4R- and 3R-tau isoform aggregates (AD and CTL cases), highlighting heterogeneity amongst the FTLD cases and justifying the subdivision into FTLD-4R and FTLD-3R groups across all subsequent analyses.”

- Table 1: The total Tau quantification data for insoluble Tau provided in this table do not seem to match the data provided in Extended data Fig. s3. Furthermore, the table legend states that total Tau was below the detection limit of quantification in the insoluble fraction, while the legend of Fig. S3 states that PTM abundances were normalized to absolute quantity of aggregated Tau.

Reply: Thank you. We are sorry for the lack of clarity in the previous version. We now provide more precise information regarding total tau measurement:

Total tau values reported in Table 1 were originally from SureQuant absolute quantification using AQUA isotopically labelled peptides corresponding to a tau N-domain sequence. Insoluble tau N-domain was indeed near the detection limit of quantification (50 attomol, see the quantification curve in next comment), probably because of its truncated nature. This was not the case for the R-domain (MTBR), whose quantification is reported in Fig. 8c (former Extended data Fig. s3 c), and which we used to normalize insoluble tau PTMs abundance, as shown in Fig. 8b (former Extended data Fig. s3 b). We decided to normalize insoluble tau PTM data because the total amount of tau peptides acquired by MS was highly variable between tauopathies.

We now provide the total tau soluble and insoluble measurements as the sum of the MTBR (3R+4R Tau) in Table 1. It is consistent with the data provided in the Fig. 8 (former Extended data Fig. s3) and highlights the need for normalization given the much lower value observed in control versus diseased conditions.

The results section has been modified accordingly (**page 2**):

“Total tau was measured using the ~~amine~~ microtubule-binding region (MTBR) ~~terminal domain~~ (i.e., the sum of 0N, ~~1N~~3R and ~~2N~~4R tau isoforms). In brain soluble tau, no significant differences were found in absolute total tau levels. In contrast, lower quantities of aggregated tau were observed in controls compared with tauopathies.”

- Extended Data Fig. S3c: Were any of the Tau levels measured here below the limit of quantification? If so, normalization to such values is inappropriate.

Reply: We can assure the reviewer that the tau MTBR absolute quantities showed in Fig. 8c (former Extended Data Fig. S3c) were above the limit of quantification (50 attomoles, see the quantification curve here below). This is now pointed out in:

Subjects and samples characteristics section (page 2): *“Total tau was measured using the ~~amine~~ microtubule-binding region (MTBR) ~~terminal domain~~ (i.e., the sum of 0N, ~~1N~~3R and ~~2N~~4R tau isoforms). In brain soluble tau, no significant differences were found in absolute total tau levels. In contrast, lower quantities of aggregated tau were observed in controls compared with tauopathies.”*

Table1 legend (page 13): “The absolute quantities of total tau measured using the tau MTBR (3R+4R isoforms, quantified by the Thermo Fisher Scientific SureQUANT method) in soluble brain extracts did not differ between conditions, but was significantly elevated in disease groups compared to controls in insoluble tau (limit of quantification: 50 attomoles).”

- Discussion: Tau PTMs in earlier disease stages, across brain regions and in soluble fractions have been assessed using antibody-based approaches (Ercan et al, Acta Neuropath Comm 2019). The statements in the bottom paragraph of page 8 may be modified by including this study.

Reply: Thank you. We have now included the reference in the Discussion (page 13):

“Our data is therefore not informative for early diagnosis of (preclinical) tauopathy. Future mass spectrometry work could investigate earlier cases and different brain regions from AD and non-AD tauopathies to evaluate the spatial and temporal progression of PTMs in the brain. To date, such a regional studies ~~has~~ have been carried out only by imaging ~~of~~ of aggregated tau from AD brains with different Braak stages⁴⁹ or by antibody-based study of oligomeric and detergent-soluble tau from controls and AD brains at different Braak stages⁵⁰.”

- Will the MS data generated in this study be made available through a repository?

Reply: The untargeted MS proteomic data (PTMs) were made available for Reviewers via Proteome Exchange Pride repository (see the reviewer account details here below) and will be made freely accessible to the scientific community, should our manuscript be accepted for publication.

Targeted MS proteomic (SureQuant absolute quantification) raw quantification data will be made accessible upon request due to incompatibilities with the template of the approved and recommended data repositories. The corresponding quantification results are available as tables in the supplementary information file.

Project Name: Specific post-translational modifications of the soluble tau protein distinguish between Alzheimer’s disease, 4R-, and 3R-tauopathies.

Project accession: PXD038901

Project DOI: 10.6019/PXD038901

Reviewer account details:

Username: reviewer_pxd038901@ebi.ac.uk

Password: 9yJQSCmD

- Figure 1a+b: please label the y-axis in the left panels.

Reply: Thank you. The y-axis in the left panels of Fig 1a+b has been labelled.

- Both MTBR and MTBD are used to describe the microtubule binding region. Please standardize.

Reply: Thank you. The term MTBR was standardized throughout the manuscript and the term MTBD has been removed from the manuscript.

Reviewer #2 (Remarks to the Author):

In the present manuscript, Zola, Balty et al use mass spectrometry on post-mortem tissue extracts to identify post-translational modifications to tau associated with tauopathies. This is an area of intense investigation in the field, but the present manuscript provides two areas of novelty: 1) highlighting differences between AD and non-AD tauopathies and 2) identifying specific PTMs in soluble and insoluble fractions. The authors show that 3R/4R levels are altered primarily in insoluble and not soluble fractions and identify some 4R-specific PTMs such as Ub-K369. The authors postulate that future work could attempt to identify these PTMs in biofluids such as plasma and CSF as potential disease biomarkers. Overall, the manuscript is interesting, novel and timely. Some points for revision:

- In the introduction, for clarity it is worth explaining that AD is a secondary tauopathy

Reply: Thank you. The introductory paragraph (page 1) has been modified:

*Tauopathies are a group of neurodegenerative diseases characterized by the accumulation of pathologically misfolded tau protein. **Albeit being classified as a secondary tauopathy due to the association of tau pathology with amyloidosis, Alzheimer's disease (AD) is the most prevalent tauopathy amongst at least twenty others^{1,2}. Furthermore, it has been well established that tau pathology correlates more strongly with AD cognitive impairment than amyloidosis³(Nelson et al., J Neuropathol Exp Neurol, 2012). The definite differential diagnosis of tauopathies is based on post mortem neuropathological examination of the brain⁴.***

- The most common 4R tauopathy is PSP, but this disease is not mentioned in the manuscript - this should be included in the introduction, and I also think the omission of PSP has an impact on data interpretation detailed below

Reply: Thank you. We now mention PSP (Page 1) and agree that it would have been interesting to include and assess PSP cases. However, unfortunately we do not have access to PSP material. Moreover, Including PSP samples would mean performing all the analyses over again in parallel to make valid comparison between disease conditions.

- The statement "...while fronto-temporal lobe degeneration (FTLD) cases due to tau pathology are always either 4R-only or 3R-only tauopathies" is untrue, FTLD-tau can also have pathology comprised of both 3R and 4R isoforms (for example FTLD linked to MAPT mutations such as R406W have NFTs similar to in AD with both 3R and 4R isoforms).

Reply: We thank the reviewer for his factual comment. The corresponding statement has been updated as follows (page 2):

*In AD brain, a combination of 4R- and 3R-tau isoforms is observed in aggregates while corticobasal degeneration (CBD) is characterized by aggregates containing 4R-tau only. In Pick's disease (PiD), only 3R-tau isoforms are found, while fronto-temporal lobe degeneration (FTLD) cases due to tau pathology are ~~always~~ **often** either 4R-only or 3R-only tauopathies^{2,7}.*

- The authors use a group of FTLD-tau cases including at least some MAPT mutation carriers. I think it is important to detail these - and the specific mutations - in the table as the mutations will impact on tau isoform deposition. For example, if mutations impacting tau splicing were included these are highly likely to impact on tau in the soluble fraction too.

Reply: Thank you. We have specified that the subjects 35 and 37 from the "FTLD-4R" group (High 4R/3R tau isoform ratio) were carrying the MAPT P301L mutation, in the manuscript's Table 1 result description (page 3) as follows:

“The insoluble tau 4R/3R ratio classification (Fig. 1a) separated subjects in three groups, namely 4R-exclusive (all five CBD and four FTLN cases, including two with the P301L MAPT mutation (subjects 35 and 37)), 3R-exclusive (all five PiD and six FTLN cases) or both 4R- and 3R-tau isoform aggregates (AD and CTL cases), highlighting heterogeneity amongst the FTLN cases and justifying the subdivision into FTLN-4R and FTLN-3R groups across all subsequent analyses.”

No other mutation has been reported in the FTLN cases of the study. Although the P301L mutation, located at the splicing-sensitive second microtubule binding repeat, is known to exclusively induce 4R- tau aggregates (as confirmed by our work), it did not seem to obviously impact 4R/3R ratios when comparing tau in soluble fractions of mutated and non-mutated FTLN-4R subjects, as shown in Fig. 1b (MAPT P301L mutation FTLN carriers have been highlighted with star symbols).

• **Western blots should be included to confirm that the soluble/insoluble fractionation preps have been successful - ideally for each disease group, as it is likely that different tau banding patterns will be observed (including truncated tau)- but at a minimum for control and AD to show successful fractionation**

Reply: We thank the reviewer for his valuable suggestion. We included a **Fig. 7** (see also below) issued from a western blot analysis of total homogenate-sarkosyl (H), soluble (S) and insoluble (P) fractions for 2 subjects of each studied group (asterisks indicating a MAPT P301L mutation carrier), using an antibody recognizing a stable C-terminal present in both soluble and aggregated forms (epitope surrounding D₄₃₀) to gather the most exhaustive signal for tau. Please also find the figure at the end of the comments.

Indeed, we observed different tau banding patterns between H, S and P in the different subject's groups. Moreover, compared to control, tau signal reached high molecular levels in homogenate and soluble fractions, suggesting highly modified tau species. Tau signal in aggregates were mostly absent in controls and for all disease groups, the signal was higher than in homogenate and soluble fraction for the same total protein input (5 µg).

Successful fractionation is demonstrated by the tau signal intensity decrease in soluble fraction compared to total-homogenate in all disease groups while unchanged in controls (where no aggregate is detected), in parallel with a constant signal in both fractions for vinculin as loading control.

• **The findings of 4R specific modifications are very interesting. The authors should include some PSP cases, as it is important to know whether these changes are common across 4R tauopathies or if they might be able to distinguish between FTLN/CBD/PSP**

Reply: As mentioned above, we agree that including PSP cases would have permitted to determine whether PTMs were able to distinguish between different types of 4R-tauopathies. However, the primary objective of our work was to provide a proof-of-concept that soluble tau PTMs were able to distinguish between AD, 3R-, and 4R-tauopathies, which could be achieved with CBD and FTLN-4R as representative cases of 4R-tauopathies.

Editorial Note: Part of the figure below has been redacted to remove third-party material where no permission to publish could be obtained.

Annex to the Reviewer #2 comment n°5:

For the facility of the reviewer, we have reproduced here the new **Fig. 7**.

Fig. 7 | Soluble and insoluble tau fractionation from total brain homogenate. To validate ultracentrifugation-based Tau fractionation from sarkosyl lysates, 5µg of proteins from each fraction (H, total sarkosyl homogenate; S, soluble; P, insoluble pellet) from two subjects per tauopathy were analysed by western blotting using an antibody recognizing a stable epitope in Tau C-terminal domain. Vinculin was used as loading control for H and S fractions. Asterisks indicate a MAPT P301L mutation carrier.

REVIEWERS' COMMENTS

Reviewer #1 (Remarks to the Author):

The authors provided a revised version of their manuscript in which they appropriately address all the points I previously raised. Unfortunately some formatting errors have occurred, and I would thus like to ask the authors to thoroughly review the figure numbers cited in the text- they do not always match the correct figure (i.e. page 3: Figure 8 is cited for Western blots, but Figure 7 is the one showing Western blots). Some referencing errors also seem to have occurred ('Reference source not found'). After correction of these errors, I have no more concerns regarding publication.

Reviewer #2 (Remarks to the Author):

The authors have address most of my comments and I find this version of the manuscript much improved. Regarding the inclusion of PSP cases, if the authors do not have access to PSP material perhaps they could also expand on this in the discussion and whether future work should aim to identify PTMs that distinguish the tauopathies.

ANSWERS TO THE REVIEWERS' COMMENTS

We thank the reviewers for their constructive and helpful comments. Please find below our answers to each of the points raised by the reviewers. Our responses are indicated in red below. Copy-pasted sections from the original manuscript are in italics with changes in red.

Reviewer #1 (Remarks to the Author):

The authors provided a revised version of their manuscript in which they appropriately address all the points I previously raised. Unfortunately some formatting errors have occurred, and I would thus like to ask the authors to thoroughly review the figure numbers cited in the text- they do not always match the correct figure (i.e. page 3: Figure 8 is cited for Western blots, but Figure 7 is the one showing Western blots). Some referencing errors also seem to have occurred ('Reference source not found').

After correction of these errors, I have no more concerns regarding publication.

We thank the reviewers for his comment. We updated the manuscript following the guidance of the Editor author check list, all the errors mentioned here above have been corrected.

Reviewer #2 (Remarks to the Author):

The authors have address most of my comments and I find this version of the manuscript much improved. Regarding the inclusion of PSP cases, if the authors do not have access to PSP material perhaps they could also expand on this in the discussion and whether future work should aim to identify PTMs that distinguish the tauopathies.

We thank the reviewers for his recommendations. We added the perspective of this work replication in the discussion as following:

“Also, we limited our study to one exemplative pathology of “4R and 3R” (AD), “4R” (CBD), “3R” (PiD) and “4R or 3R” (FTLD). Tauopathies include more than twenty different diseases, and structural differences can be found in aggregates including the same type of tau isoform (e.g., in CBD versus progressive supranuclear palsy)²⁶. Therefore, it would be worth characterizing tau PTMs in additional tauopathies to confirm whether the observed PTMs are specific for the type of isoforms found in aggregates or whether those PTMs are rather specific for the diseases that we have selected. Specifically, future work should determine whether Ub-K369 and Ub-K343 are specific for CBD or also observed in other 4R tauopathies such as PSP.”